# Sequence deeper without sequencing more: Bayesian resolution of ambiguously mapped reads

Rohan N. Shah[1,2]*, Alexander J. Ruthenburg[2,3]*

**1** Pritzker School of Medicine, Division of the Biological Sciences, The University of Chicago, Chicago, Illinois, United States of America, **2** Department of Molecular Biology and Cell Genetics, Division of the Biological Sciences, The University of Chicago, Chicago, Illinois, United States of America, **3** Department of Biochemistry and Molecular Biology, Division of the Biological Sciences, The University of Chicago, Chicago, Illinois, United States of America

* rohanshah@uchicago.edu (RNS); aruthenburg@uchicago.edu (AJR)

## Abstract

Next-generation sequencing (NGS) has transformed molecular biology and contributed to many seminal insights into genomic regulation and function. Apart from whole-genome sequencing, an NGS workflow involves alignment of the sequencing reads to the genome of study, after which the resulting alignments can be used for downstream analyses. However, alignment is complicated by the repetitive sequences; many reads align to more than one genomic locus, with 15–30% of the genome not being uniquely mappable by short-read NGS. This problem is typically addressed by discarding reads that do not uniquely map to the genome, but this practice can lead to systematic distortion of the data. Previous studies that developed methods for handling ambiguously mapped reads were often of limited applicability or were computationally intensive, hindering their broader usage. In this work, we present SmartMap: an algorithm that augments industry-standard aligners to enable usage of ambiguously mapped reads by assigning weights to each alignment with Bayesian analysis of the read distribution and alignment quality. SmartMap is computationally efficient, utilizing far fewer weighting iterations than previously thought necessary to process alignments and, as such, analyzing more than a billion alignments of NGS reads in approximately one hour on a desktop PC. By applying SmartMap to peak-type NGS data, including MNase-seq, ChIP-seq, and ATAC-seq in three organisms, we can increase read depth by up to 53% and increase the mapped proportion of the genome by up to 18% compared to analyses utilizing only uniquely mapped reads. We further show that SmartMap enables the analysis of more than 140,000 repetitive elements that could not be analyzed by traditional ChIP-seq workflows, and we utilize this method to gain insight into the epigenetic regulation of different classes of repetitive elements. These data emphasize both the dangers of discarding ambiguously mapped reads and their power for driving biological discovery.

**Data Availability Statement:** Software written as part of this work are available at https://github.com/shah-rohan/SmartMap for download. The tools included at that repository are the SmartMapPrep script, the SmartMapRNAPrep

script, and the SmartMap program. The SmartMapPrep software is used to streamline the alignment, filtering, and processing of reads to enable their use in the SmartMap software. The SmartMapRNAPrep software is used to do the same, except for strand-specific applications. The SmartMap software is used to conduct the iterative Bayesian reweighting algorithm described above and yields a gzipped BEDGRAPH file of the genome coverage of map weights. In addition, these tools are all available through Bioconda at http://bioconda.github.io/recipes/smartmap/README.html. Detailed instructions for installation and use are available at https://shah-rohan.github.io/SmartMap. All ICeChIP-seq and MNase-seq data are available at GEO under accession numbers GSE60378 and GSE103543. All RNA-seq and ATAC-seq data are available at https://www.encodeproject.org/ from the ENCODE Project under experiment accession numbers ENCSR000AEL and ENCSR483RKN, respectively. The simulated data and analysis workflow for both simulated and biological data are available on Zenodo at https://zenodo.org/record/4586639, with detailed instructions provided both in that Zenodo repository and on Github at https://shah-rohan.github.io/SmartMap/analysis.html. Simulated FASTQ files can be found on Zenodo at https://zenodo.org/record/4584103.

**Funding:** This study was supported by the National Institutes of Health (https://www.nih.gov/) under award numbers R01-GM115945-05 and R01-HL148719 to A.J.R. and T32-HD007009-45 for training grant funding provided to R.N.S. The funders had no role in study design, data collection and analysis, decision to publish, or preparation of the manuscript.

**Competing interests:** The authors have declared that no competing interests exist.

## Author summary

Next-generation sequencing allows researchers to efficiently determine the sequences of hundreds of millions of short DNA fragments from an experiment. Many experiments use next-generation sequencing to count nucleic acid molecules in a population by sequencing small fragments of them and assigning them to different genomic features. To find the origins of those fragments, the corresponding sequences are aligned to the genome; these alignments can then be used in downstream analyses. However, this alignment process is complicated by the fact that the genome has many highly similar and repetitive sequences, making it difficult or impossible to unambiguously assign some sequences to a single genomic location. The common "solution" to this problem is to discard those sequencing reads that do not align to a single site; however, this can lead to significant biases and will hide an important part of the genome. To address this problem, we have developed SmartMap, which serves to process and appropriately weight the alignments of reads that map to more than one genomic location. This enables us to examine many genomic regions that were previously "invisible" to analysis and helps us draw new insights into the regulation and function of repetitive elements of the genome.

## Introduction

The impact of next-generation sequencing (NGS) on molecular biology can hardly be overstated. In a typical short-read NGS workflow, DNA fragments from an experiment are loaded onto a sequencer, which reports the sequence of 40-200bp of one end or both ends of each fragment (in single-end or paired-end sequencing, respectively) [1]. These reads/read pairs can then be aligned to the genome by one of several alignment tools, and the set of alignments can be used to compute the number of reads aligned to any given genomic locus. This genome-wide read depth dataset can then be used in downstream workflows.

Even beyond applications for whole genome sequencing, many critical methods have leveraged NGS to enable truly genome-wide biological studies. RNA sequencing (RNA-seq) has enabled quantification of gene expression [2] as well as the discovery and characterization of new elements of the transcriptome, such as enhancer RNAs [3–6] and chromatin-associated RNAs [7,8]. Chromatin immunoprecipitation coupled to NGS (ChIP-seq) has similarly become a mainstay of molecular biology, with many of the seminal works in the field relying upon this technique [9–17]. Other common techniques, including ATAC-seq [18], Hi-C [19], CUT&RUN [20], and TAB-seq [21], similarly rely on NGS and associated workflows to provide important insights into genomic regulation.

Crucially, these workflows all rely upon alignment of each read to its corresponding genomic location. However, this can be problematic when analyzing non-unique or repetitive regions of the genome, particularly given the short window of a 40-200bp sequencing read. Indeed, some estimates suggest that a majority of the human genome is comprised by repetitive elements [22–24]. Accordingly, between 15–30% of the human genome is not uniquely mappable by single-end sequencing with typical read lengths [25,26], and the genomes of other model organisms, such as *M. musculus* or *D. melanogaster*, present similar challenges [26]. Paired-end sequencing can partially improve genome mappability, but of the regions that are not uniquely mappable by single-end sequencing, 70–85% will not be resolved by paired-end sequencing [26].

Many NGS pipelines address this ambiguity by masking repetitive regions to prevent alignment of reads to more than one genomic locus or by filtering only for reads that align

unambiguously to the genome (hereafter referred to as unireads) [27]. This includes groups such as the ENCODE Consortium, whose ChIP-seq pipeline filters for uniquely mapped reads by default [28]. Indeed, in several of our past studies, we ourselves have utilized filters to exclude ambiguously mapping reads [29–31]. However, filtering out reads that map to multiple loci (hereafter referred to as multireads) sacrifices the ability to critically examine many repetitive regions of the genome, which have important roles in gene regulation [27]. Further, by definition, discarding reads reduces read depth, which makes quantitative comparisons more challenging by increasing error or the necessary sequencing depth. [27]. Given the many problems with ignoring or discarding repetitive regions or ambiguous alignments, it is critical to develop and utilize methods to appropriately analyze multireads.

To date, several studies have attempted to develop methods and algorithms to resolve multiread alignments for a variety of applications. Some have targeted their analysis methods towards RNA-seq and quantifying transcripts [2,32,33]; indeed, in recent years, there has been a sharp increase in the tools available for quantification of pre-defined genomic features in RNA-seq [34]. Others have developed tools designed for ChIP-seq or DNA-seq more broadly [35–39].

Despite the wide array of tools that have been previously developed for this problem, there are still several outstanding problems. First, several of the previously published tools (particularly for RNA-seq) focus on quantification of a distinct set of genomic features rather than generating truly genome-wide coverage maps [2,32–34,37,38], rendering them inappropriate for ChIP-seq or other unbiased/*de novo* NGS analyses. Even amongst these remaining tools for "peak type" ChIP-seq or similar analyses, several of these tools focusing on comparison to external datasets for peak calling [37,38], leaving even fewer analysis methods for a single dataset without an exogenous reference. Second, while many existing methods use alignment weighting algorithms to allocate multiread depth, there is disagreement as to the degree to which iterative reweighting is required to properly weight the multireads without over-refining the weights; some employ no iterative reweighting at all [2,39], whereas others use up to 200 reweighting cycles [35]. In addition, most of the above methods do not consider the alignment quality when resolving read ambiguity or does so in a computationally intensive manner that would likely scale poorly with the number of reads commonly obtained from modern NGS platforms [36]. Further, these tools often focused on single-end sequencing and do not make use of the intervening length information in paired-end sequencing, limiting the scope of their applicability [35]. Finally, many of these tools do not accommodate strand-specific analyses genome-wide, limiting their application to strand-independent experiments [35–38].

In this work, we seek to resolve some of these issues. We describe SmartMap: an algorithm that uses iterative Bayesian reweighting of ambiguous mappings, with assessment of alignment quality as a factor in assigning weights to each mapping. We find that SmartMap markedly increases the number of reads that can be analyzed and thereby improves counting statistics and read depth recovery at repetitive loci. This algorithm and software implementation is compatible with both paired-end and single-end sequencing, and can be used for both strand-independent and strand-specific methods employing NGS backends to generate genome-wide read depth datasets.

## Results

### Development and validation of a Bayesian multiread allocation algorithm

We initially developed our SmartMap algorithm and software for application in ChIP-seq using a set of internally calibrated ChIP-seq (ICeChIP-seq) datasets. These datasets were previously generated by our lab and, with one exception, were previously published as components

of past studies [29,30]. We chose to use ICeChIP-seq datasets because the included internal standards allow for computation of antibody specificity and for normalization to calculate the histone modification density (HMD), or the absolute proportion of nucleosomes at a given genomic locus bearing the targeted histone modification. These additional factors which we can compute using ICeChIP-seq datasets afford us additional points of quantitative comparison to assess differences between uniread and SmartMap analyses. However, this tool is not designed solely (or even primarily) for use with ICeChIP-seq datasets; the SmartMap algorithm does not make special use of the internal standards. Rather, this software is designed to be usable for NGS workflows more broadly.

The workflows for uniread analyses (typical of ChIP-seq) and our SmartMap analysis are shown in Fig 1A. For both analyses, the immunoprecipitation (IP) and MNase-seq Input sequences are aligned to the appropriate reference genome and are filtered to select for properly mapped reads in a proper pair. At that point, the two methods diverge. In the uniread analysis, which represents our published analysis pipeline for ICeChIP-seq data [29–31], any reads that don't align uniquely are discarded, and the remainder are used to compute genome-wide read depth in the IP and Input, fold-change, and (if internal standards are present) HMD.

In the SmartMap analysis, however, rather than discarding ambiguously mapped reads, we instead feed our alignments into our iterative Bayesian reweighting algorithm, outlined in Fig 1B. Our algorithm, like other alignment weighting algorithms [2,35,36,39], is motivated by the assumption that regions with more alignments are more likely to be the true source of an multiread than those with fewer alignments. In addition, like BM-Map [36], SmartMap utilizes both paired-end sequencing information and alignment quality in making these assessments. Accordingly, our tool first assigns each alignment a weight proportional to its alignment quality, computed from the alignment software output. We then iteratively reassign weights to each alignment of each read; alignments with higher alignment quality and more overlapping alignments are assigned higher weights, and those alignments with lower quality and fewer overlapping alignments are assigned lower weights (Fig 1B). After the specified number of reweighting cycles, the resulting weights are used to compute the read depth for the IP and the Input genome-wide, which can then be used to compute fold-change or, if applicable, HMD in a similar manner as the uniread analysis. For computational efficiency, we use binary-indexed (Fenwick) trees to store genomic coordinates and associated alignment weights, much like the previously described CSEM [35]. Our implementation of these binary-indexed trees is modified to enable use of paired-end sequencing reads and, if needed, operate in a strand-specific manner.

To test this method, we created a set of simulated 50bp paired-end sequencing reads from a defined set of randomly selected genomic loci (the "true origin" loci) and used the simulated dataset to conduct uniread and SmartMap analyses (Fig 2A). The read simulation tool produces reads with "sequencing" error and also includes coverage at off-target loci to better represent the noise and off-target capture inherent in a biological experiment. Notably, the simulation enabled us to obtain the true distribution of reads (the Gold Standard), allowing us to compute the error associated with each analysis method (Fig 2A). This is particularly important because we wish to avoid over-refining the multiread weights with our inferential analysis; accordingly, the Gold Standard dataset allows us to evaluate the accuracy of reweighting.

We were particularly interested in the ability of SmartMap to recover read depth at regions of differing mappability. To investigate this relationship, we used the UMAP50 score as a measure of read mappability. The UMAP50 score for a given genomic coordinate is computed as the proportion of the 50mers covering the genomic coordinate of interest that are unique in the set of all 50mers from the genome [25]. For example, if the sequences of two of the fifty

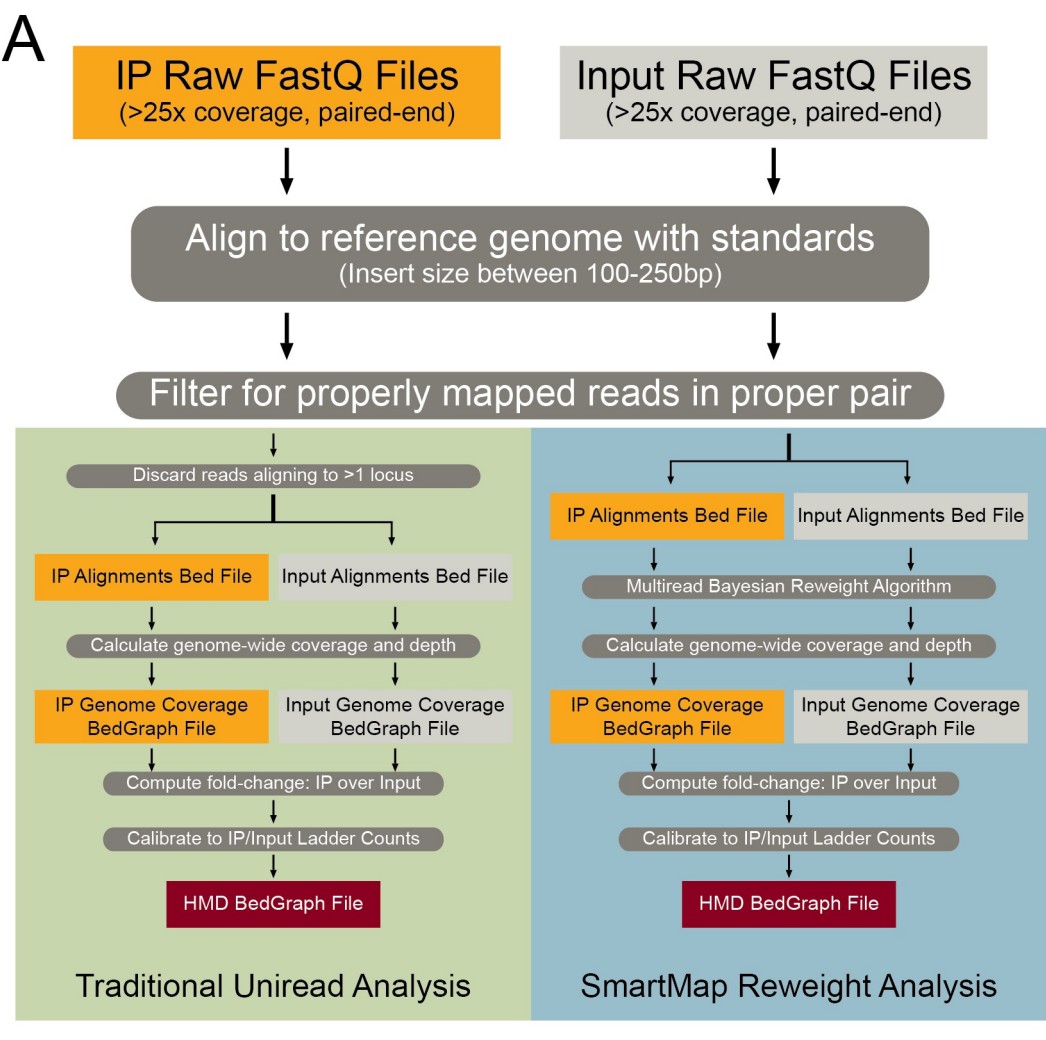

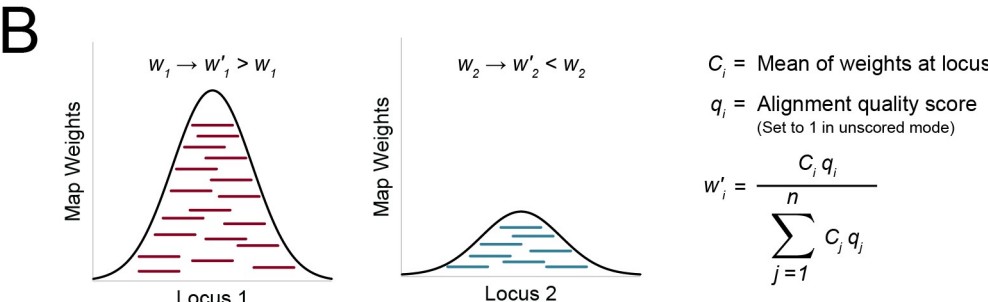

**Fig 1. Summary of the SmartMap analysis workflow and algorithm. (A)** Flowchart outlining the workflow for traditional ChIP-seq (or ICeChIP-seq) analysis [29–31] utilizing only unireads (left, green) vs. the workflow for SmartMap analysis utilizing multireads with an iterative Bayesian reweighting algorithm (right, blue). **(B)** Schematic showing the Bayesian reweighting algorithm utilized in the SmartMap analysis. Each mapping associated with a read is assigned a weight such that the weight is greater for those mappings associated with loci of greater map weight density. For more detailed description of the algorithm, see Methods.

50mers containing the genomic coordinate of interest were non-unique across the genome of study, then the UMAP50 score would be 48/50, or 0.96. As such, a genomic coordinate with a UMAP50 score closer to 1 is uniquely identified by a greater proportion of the 50mers

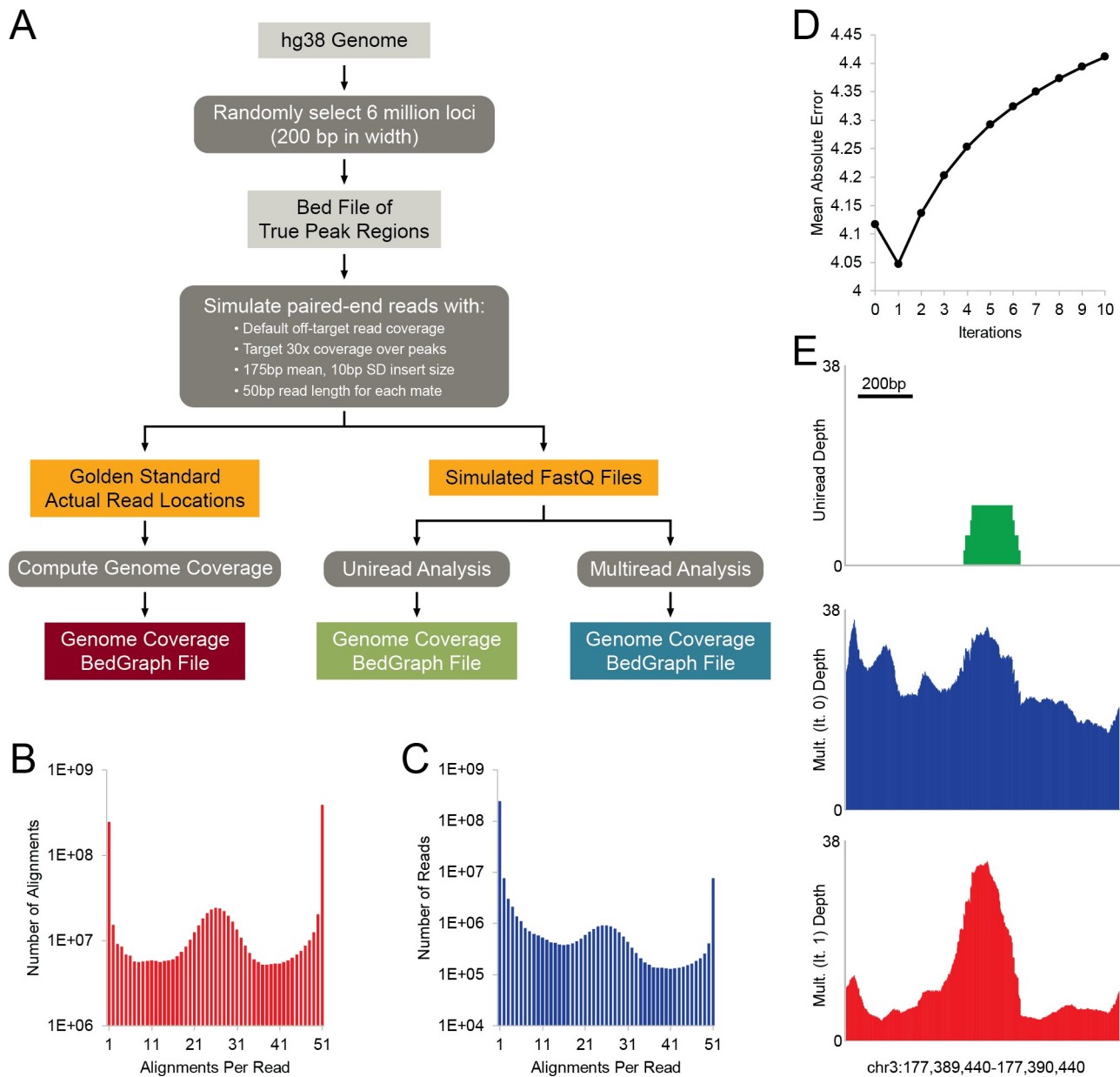

**Fig 2. Characteristics of validation dataset. (A)** Schematic outlining the workflow to validate and optimize SmartMap. A set of six million randomly selected 200bp loci were used to simulate paired end reads. The true read depth distribution was then compared to both uniread and SmartMap analyses, with each analysis conducted in both "scored" and "unscored" modes, per Methods. **(B, C)** Number of (B) alignments or (C) reads vs. number of alignments per read for the validation datasets. **(D)** Mean absolute error of read depth at true origin loci in SmartMap scored mode vs. number of reweighting iterations **(E)** Genome browser view showing the read depth in the (top) uniread, (center) SmartMap (0 iterations), and (bottom) SmartMap (1 iteration) datasets of an example locus.

spanning it than is a coordinate with a lower UMAP50 score, and a higher UMAP50 score can thus be interpreted as a more easily mappable region. Many of the true origin loci had low mappability scores (S1A Fig), with the distribution of mappability scores being similar to that of the human genome at large (S1B Fig), making this dataset useful for validating the Smart-Map algorithm.

The first step of our analysis was to align the simulated 50bp paired-end reads to the genome. We used Bowtie2 [40] with a maximum of 51 alignments reported per read and

**Table 1. Alignment statistics for the datasets used in this study.**

| | Sample | Genome | Assay | Cell Line | Unireads | Multireads | | % Increase |
|---|---|---|---|---|---|---|---|---|
| | | | | | | Analyzable | Unanalyzable | |
| | Simulated, 50bp | hg38 | Simulation | – | 245,079,644 | 34,136,124 | 7,661,326 | 13.93% |
| | Simulated, -k 101 | hg38 | Simulation | – | 244,391,815 | 35,520,969* | 6,973,053* | 14.53% |
| | Simulated, 100bp | hg38 | Simulation | – | 123,730,306 | 16,769,189 | 2,802,056 | 13.55% |
| AR7 | Input Rep. 1 | mm10* | MNAse-seq | mESC E14 | 311,090,692 | 85,018,787 | 15,184,872 | 27.33% |
| | H3K4me3 Rep. 1 | mm10* | ChIP-seq | mESC E14 | 119,014,494 | 19,662,529 | 5,603,383 | 16.52% |
| | Input Rep. 2 | mm10* | ChIP-seq | mESC E14 | 304,127,899 | 83,629,528 | 17,160,012 | 27.50% |
| | H3K4me3 Rep. 2 | mm10* | ChIP-seq | mESC E14 | 91,518,104 | 14,549,072 | 4,657,032 | 15.90% |
| AR8 | Input | dm3† | MNAse-seq | S2 | 18,678,956 | 7,117,520 | 977,776 | 38.10% |
| | H3K27me3 | dm3† | ChIP-seq | S2 | 8,855,114 | 3,249,005 | 389,227 | 36.69% |
| AR9 | Input | mm10† | MNAse-seq | mESC E14 | 488,503,092 | 131,960,514 | 26,577,525 | 27.01% |
| | H3K4me3 | mm10† | ChIP-seq | mESC E14 | 169,335,369 | 32,089,449 | 7,918,756 | 18.95% |
| | H3K9me3 | mm10† | ChIP-seq | mESC E14 | 136,008,760 | 73,118,061 | 13,012,319 | 53.76% |
| | H3K27me3 | mm10† | ChIP-seq | mESC E14 | 155,322,021 | 43,508,387 | 9,267,806 | 28.01% |
| AR16 | Input | hg38‡ | MNAse-seq | K562 | 285,996,344 | 56,595,547 | 12,902,707 | 19.79% |
| | H3K4me1 | hg38‡ | ChIP-seq | K562 | 92,422,802 | 16,475,108 | 2,434,216 | 17.83% |
| | H3K4me2 | hg38‡ | ChIP-seq | K562 | 70,987,452 | 12,931,282 | 2,558,979 | 18.22% |
| | H3K4me3 | hg38‡ | ChIP-seq | K562 | 40,483,145 | 5,488,996 | 803,892 | 13.56% |
| AR17 | Input | hg38‡ | MNAse-seq | K562 | 256,373,920 | 48,634,887 | 11,216,500 | 18.97% |
| | H3K9me3 | hg38‡ | ChIP-seq | K562 | 193,011,406 | 40,618,196 | 10,337,413 | 21.04% |
| | H3K27me3 | hg38‡ | ChIP-seq | K562 | 173,915,939 | 32,770,085 | 7,107,199 | 18.84% |
| ENCODE | Snyder Rep. 1 | hg38 | ATAC-seq | K562 | 32,995,935 | 6,484,894 | 299,834 | 19.65% |
| | Snyder Rep. 2 | hg38 | ATAC-seq | K562 | 24,414,870 | 4,210,386 | 149,154 | 17.25% |
| | Gingeras Rep. 1 | hg38§ | RNA-seq | K562 | 60,184,580 | 20,651,064 | 29,231 | 34.31% |
| | Gingeras Rep. 2 | hg38§ | RNA-seq | K562 | 63,238,387 | 13,087,755 | 14,070 | 20.70% |

For all datasets, Unireads refers to the number of reads with one alignment.

For all except the "Simulated, -k 101" dataset, Analyzable Multireads refers to reads with between 2–50 alignments; Unanalyzable Multireads refers to reads with 51 reported alignments, the limit for reported alignments per read.

For the "Simulated, -k 101" dataset, Analyzable Multireads refers to reads with 2–100 alignments, and Unanalyzable Multireads refers to reads with 101 reported alignments.

% Increase: Increase in the number of analyzable reads with SmartMap analysis, computed as the number of Analyzable Multireads as a percentage of the number of Unireads.

Genome includes ICeChIP barcodes:

* Series 1.

† Series 2.

‡ Series 3.

§ Genome includes ENCODE ERCC standards.

charted the distributions of the number of alignments per read (Fig 2B and 2C). Notably, we observed that there were many reads that did not uniquely align to the genome; approximately 17.1% of the simulated reads mapped to more than one locus (Fig 2B and 2C and Table 1).

Our first goal was to determine the optimal number of iterations to use for our SmartMap analyses. To test this, we computed the mean absolute error of SmartMap read depth at the true origin loci with varying numbers of reweighting cycles, as compared to the Gold Standard read depth. Surprisingly, we found that the lowest error occurred after only one reweighting cycle (Fig 2D), with genome browser views showing refinement of peak structure (Fig 2E), which is particularly important given the importance that has been placed on peak breadth

[41]. This stands in stark contrast with previous works, which have used up to 200 iterations of reweighting [35]. Our analysis here, however, shows that may be suboptimal, suggesting that applying Bayesian alignment reweighting more than once may over-refine the data.

We wanted to explore whether these increases in mean absolute error were systematic or driven by random "overshoot" of weight at each locus. In the former case, we might expect to see that the true origin loci would either show systematic increases or decreases in read weight with greater numbers of reweighting cycles. In the latter case, we would expect that the changes to each weight might increase or decrease by too much in the initial iteration, which would present as random, relatively unbiased errors.

To distinguish between these two possibilities, we conducted two analyses. First, we computed mean error of weights at the true origin loci (S2A Fig) rather than the mean absolute error (Fig 2D). If there was a systematic erroneous increase or decrease in the average read depth of each locus, then we would observe a corresponding increase or decrease in mean error with more iterations, respectively. However, what we instead observe is that the mean error is relatively stable from iterations 2–8 (S2A Fig), suggesting that the marked increase in mean absolute error with increasing iterations is not primarily caused by systematic erroneous increases or decreases in locus weight depth. Put differently, it does not appear that the true loci are incorrectly and systematically "pulling in" or "pushing out" read depth with each reweighting cycle.

Second, we explored the possibility that the reweighting "overshoots" the weight adjustment for reads at random. If this was the case, then we would expect that the errors would increase relatively randomly, with both positive and negative errors. Indeed, this is what we observe in our analysis of mean error by iteration (S2A Fig). In addition, we would predict slowing the rate of weight adjustment with each cycle would decrease the amount of overshoot and thereby lead to a lesser increase in error. To test this, we introduced a tunable reweighting rate parameter such that the weights could be changed less with each reweighting iteration. When applying a reweighting rate of 0.25 (wherein the weights only change by 25% as much in normal SmartMap analysis), we found that the mean absolute error was markedly more stable after one iteration (S2B Fig). Indeed, after two cycles of standard SmartMap, the mean absolute error exceeds that of the dataset with no reweighting (Fig 2D); by contrast, with eight cycles of SmartMap with a reweighting rate of 0.25, the mean absolute error is considerably below that of the iteration 0 dataset and comparable to the minimum mean absolute error after one iteration (S2B Fig). This suggests that the increase in error with increasing iterations observed with standard SmartMap may be due to "overshoot" of reweighting, which compounds in magnitude with further reweighting. Interestingly, we found that the mean absolute error with one iteration of standard SmartMap analysis was on par with (and even slightly lower than) that of the slow-reweighting dataset (Figs 2D and S2B), suggesting that this potential overshoot error may not be too detrimental after only one iteration of reweighting. By command line switch, these two algorithms are both available in the SmartMap software.

After determining the optimal number of reweighting cycles, we then compared the SmartMap and uniread analyses of our simulated datasets (Fig 2A). To determine the relative impact of using alignment quality for multiread analysis, we ran SmartMap in both scored and unscored modes. All the SmartMap analyses had greater read depth (and were closer to the Gold Standard dataset) at true origin loci than the corresponding uniread analyses (Fig 3A). Interestingly, the increases in read depth were not uniform across the entire set of loci; indeed, approximately 70% of the true origin loci saw no excess read depth, defined as the difference between SmartMap and uniread read depths (Fig 3B). This is similarly observed in the QQ plot comparing uniread and SmartMap analyses; a shoulder is seen at low uniread depth, with the plot converging onto a slope of unity at higher read depths (Figs 3C and S2C), suggesting

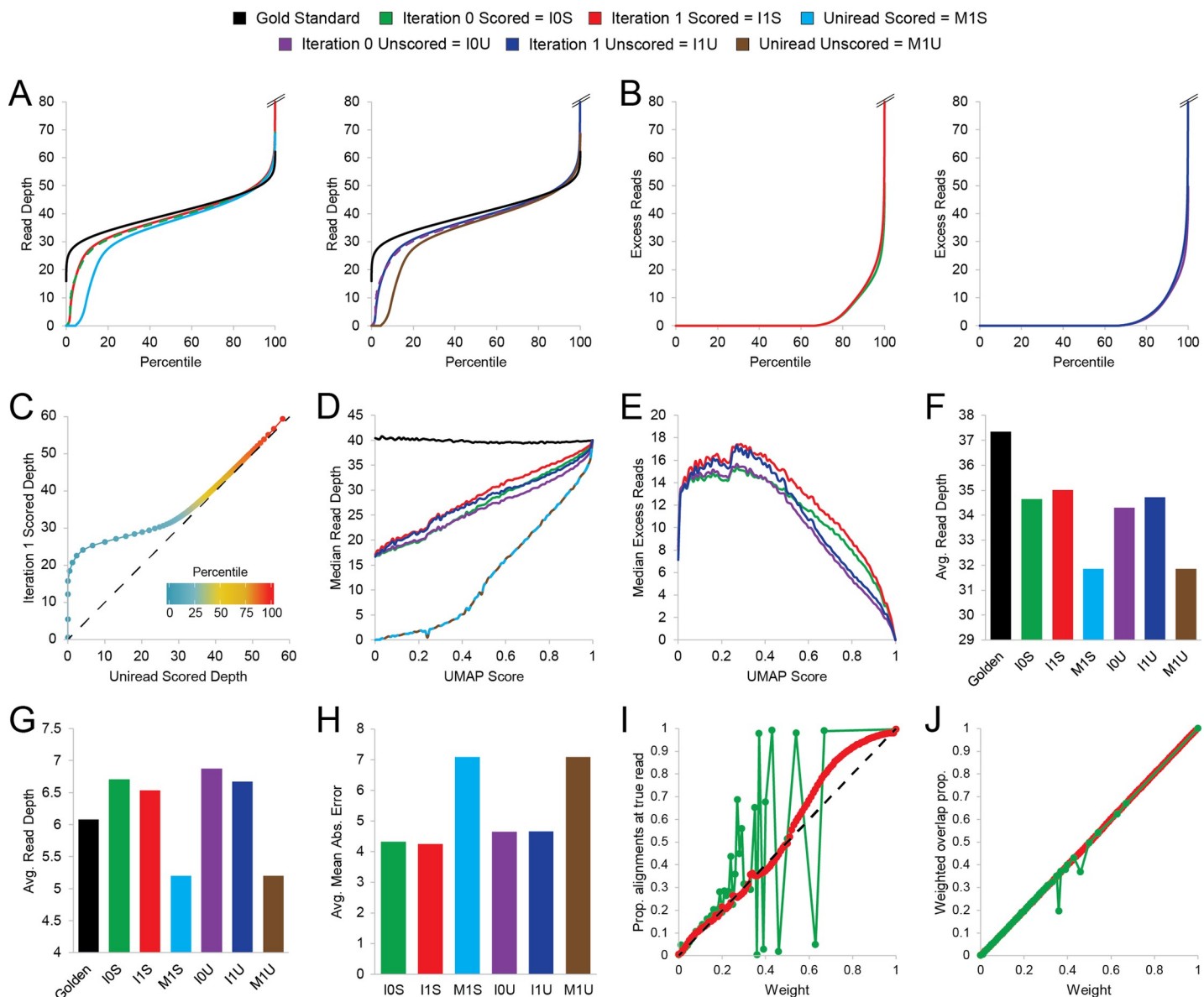

**Fig 3. SmartMap and uniread analyses of the validation dataset.** Iteration 0 and iteration 1 refer to SmartMap analysis with 0 and 1 iterations of reweighting, respectively. Scored and unscored refer to whether alignment score was considered in analysis, per Methods. Dashed lines are presented for readability of overlapping curves rather than discontinuities in data throughout this figure. **(A)** Quantile plot of read depth at the true origin loci, with Gold Standard dataset and analysis conducted in (left) scored mode or (right) unscored mode. **(B)** Quantile plot of excess read depth in SmartMap datasets relative to corresponding uniread dataset at true origin loci in (left) scored mode and (right) unscored mode. **(C)** QQ plot of read depth at true origin loci in the SmartMap (1 iteration) scored dataset vs. uniread scored dataset. Color scale represents percentile of each point, from 1st to 99th percentiles. **(D-E)** Median (D) read depth or (E) excess read depth vs. mappability score (UMAP50) [25] of the true origin loci. **(F-G)** Average read depth (F) at true origin loci and (G) outside true origin loci. **(H)** Mean absolute error of read depth at true origin loci for each dataset, with Gold Standard as the reference point. **(I)** Mean proportion of alignments intersecting with the true read of origin for each weight after SmartMap with no reweighting (green) and one iteration of reweighting (red) in scored mode. Dashed line represents line with slope of unity. **(J)** Mean weighted overlap proportion score between alignments intersecting the true read of origin and the true read locus for each weight after SmartMap with no reweighting (green) and one iteration of reweighting (red) in scored mode. Weighted overlap proportion score is meant to represent the proportion of a read's weight that maps to the correct location due to a particular alignment and is computed as a weighted geometric mean of the proportion of the alignment covered by the true read and the proportion of the true read covered by the alignment.

that the gains in read depth were primarily at regions of low uniread depth. While SmartMap does not fully recover the read depth of the Gold Standard at the low end of the QQ plot, the shoulder is nonetheless much less prominent than with the uniread analysis (S2D Fig),

indicating considerably greater depth recovery. Consistent with that observation, the uniread analyses and the SmartMap analyses both performed well at highly mappable regions, with read depths approximately at the level of the Gold Standard (Fig 3D). However, at regions of lower mappability, the SmartMap analyses recovered a markedly greater proportion of the read depth than did the uniread analyses (Fig 3D and 3E). As expected from prior analyses (Fig 2D), the SmartMap analyses with one iteration of reweighting recovered greater read depth than those with no reweighting (Fig 3D and 3E). Importantly, though they performed similarly at regions of lower mappability, the SmartMap scored analyses recovered greater read depth than their unscored counterparts at regions with moderate mappability scores (Fig 3D and 3E).

Genome-wide, SmartMap analyses had lower on-target read depth than the Gold Standard dataset but were still able to recover greater depth at the on-target loci than corresponding uniread analyses (Fig 3F). Similarly, the SmartMap analyses had marginally higher off-target read depth than the Gold Standard and uniread datasets (Fig 3G); however, the increased off-target depth relative to uniread datasets can be explained by the overall lower read depth in the uniread datasets (S3A Fig). Consistent with the notion that improved priors enhance Bayesian predictions, the unscored SmartMap analyses had lower on-target and higher off-target read depth than the corresponding scored analyses (Fig 3F and 3G), and the no-iteration SmartMap analyses had similarly lower on-target and higher off-target read depth than their one-iteration counterparts.

As another metric to evaluate each analysis, we conducted MACS2 peak calling on each dataset and assessed the degree to which they overlap. The SmartMap analyses had similar (albeit slightly higher) base pair coverage with called peaks relative to the Gold Standard dataset and considerably higher coverage on called peaks than the uniread analyses (S3B Fig), consistent with the genome-browser views that suggest a similar pattern of peak boundary sharpening (S1 Fig). As a measure of sensitivity, we computed the proportion of the Gold Standard peaks that were covered by SmartMap or uniread peaks (S3C Fig). Conversely, to measure specificity, we computed the proportion of SmartMap or uniread peaks that were covered by Gold Standard peaks (S3D Fig). As expected, there was considerably lower coverage by the uniread datasets than the SmartMap datasets, and the one-iteration SmartMap analyses had very slightly lower coverage over the Gold Standard peaks than the no-iteration analyses (S3C Fig). However, the one-iteration analyses were better-covered by Gold Standard peaks than were their no-iteration counterparts (S3D Fig). Together, these data suggest that SmartMap analyses with one iteration of reweighting have a marked increase in specificity relative to the no-iteration analyses at the expense of a slight decrease in sensitivity.

We then evaluated the overall mean absolute error of read depth at the true origin loci relative to Gold Standard. The uniread analyses had the highest average mean absolute error, with all SmartMap analyses outperforming all uniread analyses (Fig 3H). The scored SmartMap analyses also all had lower error than did the unscored analyses, and the one-iteration SmartMap analyses slightly outperformed the no-iteration analyses (Fig 3H). The error in all datasets tended to primarily be concentrated at regions of lower mappability (S3E Fig). Interestingly, though SmartMap with one iteration had lower mean absolute error overall, the no-iteration modality had slightly lower mean absolute error at true origin loci of lower read depth (Figs 3H and S3F). The reason for this difference is not clear; across all read depth classes, the one-iteration analyses had slightly less negative mean error, suggesting that there wasn't a large-scale difference in over- or underweighting after iteration as a function of read depth (S3G Fig). With that said, we feel it is important to contextualize these results; these differences between the no-iteration and one-iteration analyses are small in magnitude and are comparatively dwarfed by the differences between SmartMap and uniread analyses (Figs 3H and S3E–

S3G). Accordingly, though there may be small differences between the no-iteration and one-iteration SmartMap analyses, the one-iteration analyses still performed better in aggregate, and both of these scored SmartMap analyses consistently outperformed their unscored or uniread counterparts.

The above analyses all focused on validating SmartMap from the perspective of the total read depth across a set of genomic intervals. However, given that we had a Gold Standard dataset listing the true positions of each read, we also wished to evaluate whether our reweighting method could improve the estimated of the probability that an alignment was properly mapped–and, by proxy, improve the MAPQ score estimate for each alignment. Without reweighting, the probability of correct alignment ranged from 0–0.67 and 1, with no alignments with correct alignment probability between 0.67 and 1. One iteration of SmartMap reweighting expanded the spectrum of possible alignment weights to the full range of 0–1. Without reweighting, the weight of alignments did not correlate well with the proportion of alignment intersecting the true genomic position, with many large deviations seen from linearity (Fig 3I). By contrast, though one iteration of reweighting still showed some deviations from linearity by this analysis, the weight of alignments more closely concorded with the proportion of the alignments intersecting the true read origin (Fig 3I). This suggests that by this measure, SmartMap reweighting improved the estimates of the probability that the alignment intersects with the true genomic position of the corresponding read. Similarly, we compared the weighted proportion of overlap between the true read positions and any intersecting alignments as a function of alignment weight. This is meant to represent the proportion of a read's weight that is mapped to the correct location due to a given alignment and incorporates both the confidence of the alignment selection (i.e. the weight) and the overlap of the alignment with the true origin of the read. In both the no-reweighting and one iteration analyses, the overlap proportion score was closely linearly related to the alignment weight, though the reweighted analysis showed a slightly smoother curve with fewer marked deviations from linearity (Fig 3J). This is roughly expected, as the overlap proportion score is itself a function of weight; however, this analysis is comforting insofar as it shows that the SmartMap reweighting does not markedly inflate or deflate the expected weight contribution of a given alignment to a proper intersection with the true origin. Similarly, we find that the unweighted overlap proportion of alignments with the true origin of the read is roughly constant near one for both the no-iteration and one-iteration datasets, though again, the one-iteration SmartMap analysis reduces the deviations from this level (S3H Fig). These analyses suggested that in addition to improving measurement of read depths in aggregate, the SmartMap reweighting procedure can also improve the estimates of correct alignment for individual reads and alignments.

The biological ChIP-seq and MNase-seq datasets presented in the remainder of this work used 50bp read lengths or shorter, which is why we used 50bp read lengths for our simulated dataset. However, in recent years, 100bp read lengths have become commonplace, and indeed, the ENCODE datasets we present later in this work employed paired-end 100bp NGS. As such, we examined the degree to which SmartMap can improve recovery of sequencing depth with longer reads by conducting a similar analysis as the above with a similarly simulated dataset employing 100bp paired-end reads. For facile comparison to the other figures and analyses in this work, we have continued to use the UMAP50 score as our mappability score. This choice is in spite of the fact that UMAP50 measures mappability by 50mers rather than 100mers and will thus underestimate mappability by 100bp reads. Because we are using this metric, regions with lower mappability scores will often be more easily mapped than the score would indicate, blunting differences between SmartMap and uniread analyses. As such, our analyses using the UMAP50 score will offer a very conservative view at the impact of SmartMap analysis on read depth recovery and error.

Despite this conservative choice of mappability score, we still see that SmartMap analysis improves sequencing depth recovery nearly as well with 100bp reads as it does with 50bp reads. The simulated dataset with 50bp reads shows a 13.9% increase in analyzable reads due to the high number of multireads (2B and Fig 2C and Table 1); the simulation with 100bp reads shows a 13.6% increase in analyzable reads and a similar proportion of multireads (S4A and S4B Fig and Table 1). Along the same lines, the two simulations increase read depth over similar proportions of the genome (Table 2). Over the true origin loci, much like the 50bp simulation, the 100bp simulated dataset shows an increase in read depth on quantile plots (S4C Fig) under SmartMap analysis, with this increase in read depth primarily occurring at regions of low UMAP50 mappability score (S4D Fig), conservative though this measurement of mappability is. Much like the 50bp simulated datasets, the increases in read depth under SmartMap

**Table 2. Analysis of reads across genomic windows.**

| | Sample | Genome | Assay | Regions | Regions with reads in: | | % Reg. Inc. | % Inc. Reg. |
|---|---|---|---|---|---|---|---|---|
| | | | | | Uniread | SmartMap | | |
| | Simulated 50bp | hg38 | Simulation | 15,498,848 | 10,486,482 | 11,994,872 | 28.74% | 14.38% |
| | Simulated, -k 101 | hg38 | Simulation | 15,498,848 | 10,463,337 | 12,012,046 | 29.07% | 14.80% |
| | Simulated 100bp | hg38 | Simulation | 15,498,848 | 9,956,521 | 11,475,027 | 32.77% | 15.25% |
| AR7 | Input Rep. 1 | mm10* | MNAse-seq | 13,654,309 | 12,129,867 | 13,243,873 | 33.49% | 9.18% |
| | H3K4me3 Rep. 1 | mm10* | ChIP-seq | 13,654,309 | 11,329,858 | 12,999,672 | 27.21% | 14.74% |
| | Input Rep. 2 | mm10* | ChIP-seq | 13,654,309 | 12,115,174 | 13,242,243 | 31.96% | 9.30% |
| | H3K4me3 Rep. 2 | mm10* | ChIP-seq | 13,654,309 | 10,952,182 | 12,750,113 | 27.25% | 16.42% |
| AR8 | Input | dm3† | MNAse-seq | 698,569 | 617,424 | 681,457 | 17.92% | 10.37% |
| | H3K27me3 | dm3† | ChIP-seq | 698,569 | 612,050 | 680,193 | 17.39% | 11.13% |
| AR9 | Input | mm10† | MNAse-seq | 13,654,309 | 12,214,070 | 13,245,567 | 35.60% | 8.45% |
| | H3K4me3 | mm10† | ChIP-seq | 13,654,309 | 11,775,058 | 13,208,421 | 30.83% | 12.17% |
| | H3K9me3 | mm10† | ChIP-seq | 13,654,309 | 12,027,438 | 13,245,567 | 32.11% | 10.04% |
| | H3K27me3 | mm10† | ChIP-seq | 13,654,309 | 12,012,091 | 13,237,339 | 31.99% | 10.20% |
| AR16 | Input | hg38‡ | MNAse-seq | 15,498,848 | 13,879,635 | 14,629,457 | 34.59% | 5.40% |
| | H3K4me1 | hg38‡ | ChIP-seq | 15,498,848 | 13,310,801 | 14,423,602 | 31.07% | 8.36% |
| | H3K4me2 | hg38‡ | ChIP-seq | 15,498,848 | 13,298,178 | 14,443,778 | 30.56% | 8.61% |
| | H3K4me3 | hg38‡ | ChIP-seq | 15,498,848 | 10,338,102 | 12,270,858 | 25.24% | 18.70% |
| AR17 | Input | hg38‡ | MNAse-seq | 15,498,848 | 13,896,029 | 14,634,051 | 34.56% | 5.31% |
| | H3K9me3 | hg38‡ | ChIP-seq | 15,498,848 | 13,856,547 | 14,626,552 | 34.14% | 5.56% |
| | H3K27me3 | hg38‡ | ChIP-seq | 15,498,848 | 13,803,814 | 14,618,351 | 33.66% | 5.90% |
| ENCODE | Snyder Rep. 1 | hg38 | ATAC-seq | 15,498,848 | 10,389,635 | 11,970,867 | 28.34% | 15.22% |
| | Snyder Rep. 2 | hg38 | ATAC-seq | 15,498,848 | 9,772,547 | 11,251,766 | 21.53% | 15.14% |
| | Gingeras Rep. 1 | hg38§ | RNA-seq | 41,929 | 21,755 | 25,711 | 22.85% | 18.18% |
| | Gingeras Rep. 2 | hg38§ | RNA-seq | 41,929 | 12,399 | 14,485 | 11.96% | 16.82% |

For all except the ENCODE RNA-seq datasets, analysis is conducted on 200bp genomic windows. For ENCODE RNA-seq datasets, analysis is conducted on distinct Refseq genes.

% Reg. Inc.: Percent of the total regions in the SmartMap dataset with increased read depth relative to the Uniread dataset.

% Inc. Reg.: Percent increase in the number of regions with reads in the SmartMap dataset relative to the Uniread dataset.

Genome includes ICeChIP barcodes:

* Series 1.

† Series 2.

‡ Series 3.

§ Genome includes ENCODE ERCC standards.

analysis are primarily seen at regions of low mappability and low uniread read depths; QQ plots comparing the uniread analysis with the SmartMap or Gold Standard show a shoulder at low uniread depths, with the plot converging onto a slope of unity at higher uniread depths (S4E and S4F Fig). It should be noted that, as with the 50bp simulated dataset (S2D Fig), the SmartMap dataset still fails to fully recover read depth as compared to Gold Standard with 100bp reads (S4G Fig). Nonetheless, the SmartMap analysis still shows considerably lower mean absolute error than does the uniread analysis at true origin loci (S4H Fig), with this decrease in error being particularly prominent at regions with lower UMAP50 mappability scores (S4I Fig). In total, these analyses suggest that even for datasets employing 100bp paired-end sequencing reads, multiread analysis still has nearly undiminished importance and that SmartMap can still markedly improve read depth recovery while decreasing overall error.

The above analyses all restricted Bowtie2 to report a maximum of 51 alignments for computational efficiency. Subsequently, only those reads aligning to fewer than 51 alignments were used for SmartMap analysis. However, this practice excluded more than 7 million reads (Table 1), likely including reads that map to the most highly repetitive regions of the genome. Notably, this is a restriction on alignment itself, not SmartMap; there's no reason that SmartMap would inherently be unable to handle greater numbers of alignment. Nonetheless, to evaluate the impact of this restriction on the SmartMap datasets, we reanalyzed our simulated 50bp read length dataset with a maximum of 101 alignments per read (hereafter, the k101 dataset) and compared it to the previous analysis (the k51 dataset). To our surprise, the two analyses were highly similar despite the near-doubling in the maximum-alignments threshold in the former dataset. The increase in the number of analyzable reads was nearly identical between the two analyses (S5A and S5B Fig and Table 1), with similar increases in depth over genomic windows (Table 2). At the true origin loci, the SmartMap read depths in both the k51 and k101 datasets were very similar at the level of read depth (S5C and S5D Fig). Mean absolute error relative to the Gold Standard was actually very slightly lower in the k51 dataset, though they were quite similar in magnitude compared to the uniread dataset (S5E and S5F Fig). Even specifically examining repetitive elements, read depth was very similar between the k51 and k101 SmartMap analyses at all repeats (S5G Fig), LINEs (S5H Fig), SINEs (S5I Fig), and Alu elements (S5J Fig), closely approximating the Gold Standard read depth in both cases. Accordingly, though there is still a large proportion of reads that mapped to still greater numbers of loci, we find that at the range we have tested, the SmartMap analyses are robust to differences in maximum-alignments reporting thresholds and that there is little practical difference between restricting datasets to a maximum of 51 or 101 alignments per read besides the additional time and storage space needed for the latter.

To be sure, the reweighting used for SmartMap is not without concerns. In particular, one of the potential problems for SmartMap is the existence of high-signal regions, which can show falsely high read depth in NGS experiments due to sequencing or alignment error [42]. If there are regions of falsely high weight, then those regions could be skewed by the SmartMap reweighting algorithm to report even greater weights, thus exacerbating these artifactually high signals. To assess the degree to which these regions represent an issue for SmartMap, we computed the number of genomic windows with more than 60, 70, 80, or 90 reads in our simulated datasets (Table 3). We used these benchmarks as rough thresholds for defining high-signal regions because the Gold Standard dataset had a maximum read depth across a genomic window of approximately 83 reads. Notably, the Gold Standard did not require sequencing or mapping and should thus not be susceptible to these high-signal artifacts. Unfortunately, one iteration of SmartMap reweighting did increase the proportion of high-signal regions considerably; there were fewer than 600 genomic windows with an average depth of more than 70 in the Gold Standard, Iteration 0, and Uniread datasets, compared to more than 10,000 in the

**Table 3. Analysis of high-depth regions under SmartMap analysis.**

| Dataset | Number of genomic windows with: | | | | Percent of genomic windows with: | | | |
|---|---|---|---|---|---|---|---|---|
| | >60 rds. | >70 rds. | >80 rds. | >90 rds. | >60 rds. | >70 rds. | >80 rds. | >90 rds. |
| Gold Std. | 34,468 | 463 | 1 | 0 | 0.22 | 0.0030 | $6.5 \times 10^{-6}$ | 0 |
| Iteration 0 | 26,969 | 571 | 85 | 36 | 0.17 | 0.0037 | $5.5 \times 10^{-4}$ | $2.3 \times 10^{-4}$ |
| Iteration 1 | 44,185 | 10,193 | 6,337 | 4,296 | 0.29 | 0.066 | 0.041 | 0.028 |
| Uniread | 24,344 | 321 | 1 | 0 | 0.16 | 0.0021 | $6.5 \times 10^{-6}$ | 0 |

Number of genomic windows refers to the number of 200bp genomic windows for each dataset with an average depth or average weight greater than that indicated in each column. Percent of genomic windows refers to the number of genomic windows as a percentage of the total number of 200bp genomic windows in hg38 (15,498,848). The median read depth was 10.5 and the mean read depth was 16.1 in the Gold Standard dataset.

SmartMap dataset with one reweighting cycle. It's important to note that these regions represent a very small proportion of the genome; only 0.066% of the genomic windows had more than 70 reads on average, and even fewer had more than 80 or 90 reads (Table 3), leaving considerably more than 99.9% of the genome as not having abnormally high-signal attributable to SmartMap. In contrast, almost 15% of the genome is hidden from uniread analysis (Table 2). Nonetheless, we feel it is fair to say that the reweighting algorithm used for SmartMap will increase the weights of multiread alignments at high signal regions, which can exacerbate artifactually high read depths.

Even so, on the whole, these analyses suggest that SmartMap recovers read depth at a large set of loci that would otherwise be missed by the uniread analyses and that of the SmartMap analyses, one iteration of reweighting with use of alignment scores largely outperforms the other modalities. Accordingly, for the remainder of this work, we use SmartMap analysis with one iteration in scored mode as our default SmartMap method.

## Utilizing SmartMap on MNase-seq and ChIP-seq datasets

Having validated our method on the simulated dataset, we turned to the biological samples. We deployed a total of 21 datasets derived from three different organisms for our analysis (Table 1). Of these datasets, six were control ICeChIP Inputs, generated by MNase-seq [29,30], 11 were ICeChIP-seq IP datasets, two were ATAC-seq datasets, and two were RNA-seq datasets. After alignment, the samples showed a 13–50% increase in the number of usable reads for SmartMap analysis relative to uniread (S6 and S7 Figs and Table 1).

To evaluate the impact of our algorithm on the ICeChIP-seq datasets, we first conducted SmartMap and uniread analysis on each of the Input datasets and computed the average read depth on 200bp genomic windows. As with the simulated dataset, the SmartMap analyses of the Inputs had increased read depth relative to the uniread datasets (Figs 4A and S8A), with markedly greater depth in the SmartMap analysis at windows of lower mappability (Figs 4B and S8B). Similarly, this excess read depth was not distributed across all reads, but rather, was concentrated onto 17–35% of windows (Figs 4C and S8C and Table 2), primarily at regions of lower mappability (Figs 4D and S8D). The QQ plots of the SmartMap vs. the uniread read depths showed a shoulder at low uniread depth (Figs 4E and S8E), again suggesting that the increase in read depth from the SmartMap analysis is primarily at loci where the uniread analysis performs poorly. This difference in the distributions of read depths further comments on the importance of analyzing multireads.

With our Input datasets, we could also examine the reproducibility of the MNase-seq experiments under uniread and SmartMap analyses. There were three biological replicates of Input in mESC E14 cells (AR7 Replicate 1, AR7 Replicate 2, and AR9), and two biological replicates

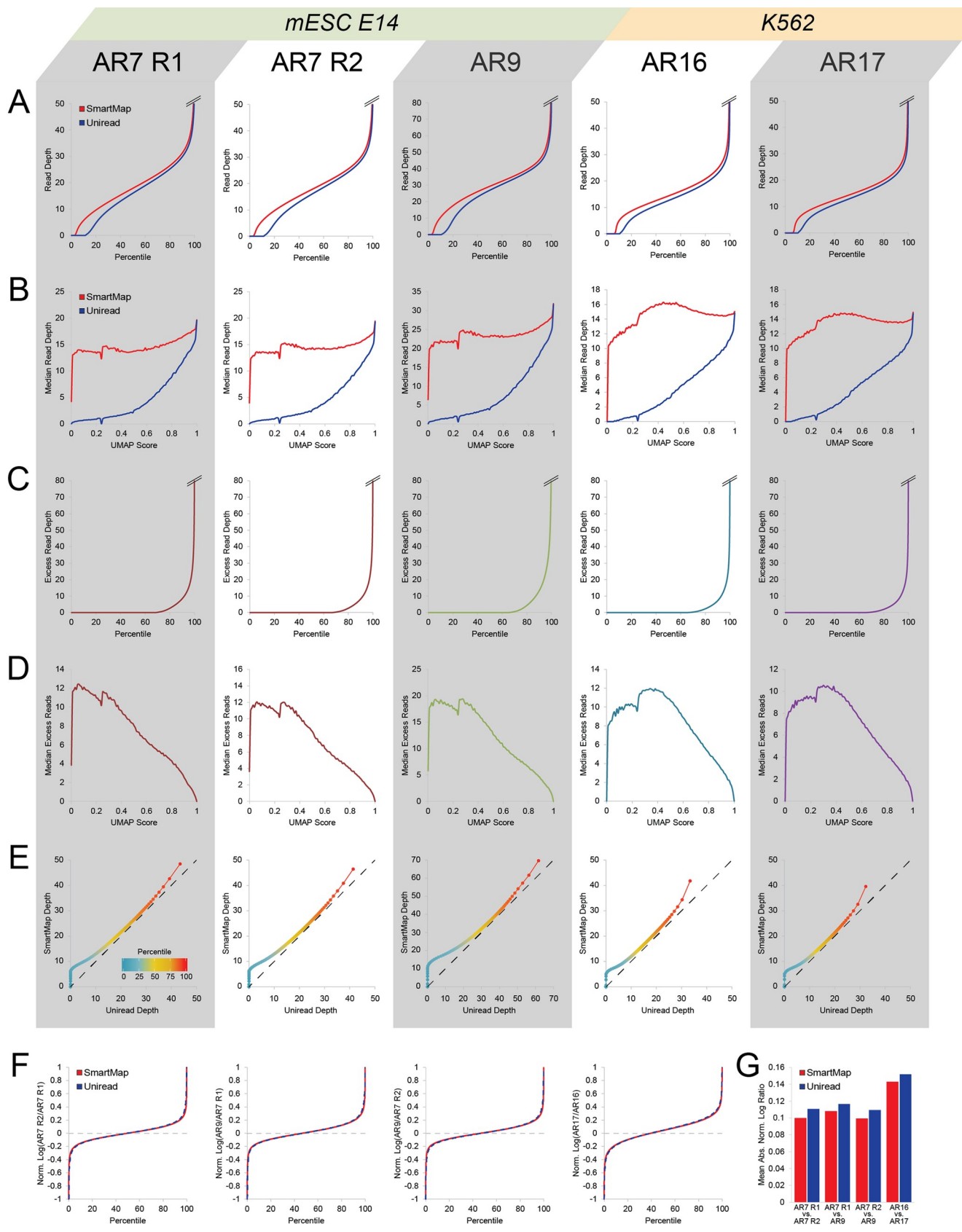

**Fig 4. SmartMap and uniread analyses of ICeChIP-seq input depth.** All analyses conducted on 200bp genomic windows for the Inputs defined in Table 2. **(A)** Quantile plot of read depth for SmartMap and uniread analyses. **(B)** Median read depth vs. mappability score (UMAP50) for SmartMap and uniread analyses. **(C)** Quantile plot of excess read depth in SmartMap relative to uniread analysis. **(D)** Median excess read depth vs. mappability score (UMAP50). **(E)** QQ plot of read depth in SmartMap vs. uniread analysis. Color scale represents percentile of each point, from $1^{st}$ to $99^{th}$ percentiles. Dashed line represents line with slope of unity. **(F)** Quantile plots of depth-normalized log ratio of read depths of biological input replicates under SmartMap and uniread analysis. Graph breaks are present on both the upper and lower ends of the graphs. **(G)** Mean absolute depth-normalized log ratio for the comparisons presented in panel F.

of Input in K562 cells (AR16 and AR17). For all loci with nonzero read depth, we computed the depth normalized log ratios of reads in a pairwise manner for biological replicates, shown as quantile plots in Fig 4F. These plots are highly similar under SmartMap and uniread analyses across all pairwise comparisons (Fig 4F). Accordingly, the average magnitudes of these ratios are similar between the two analyses–and indeed, are slightly lower in the SmartMap datasets (Fig 4G). This suggests that the two modalities show highly similar estimates of reproducibility of data between biological replicates of MNase-seq.

Having examined the Input datasets, we then used the ICeChIP-seq datasets to compute histone modification densities (HMD) across 200bp genomic windows with both uniread and SmartMap analyses. Interestingly, we noted that the mean HMD was quite similar between the SmartMap and uniread datasets across a broad range of mappability scores (Figs 5A and S9A). However, the median HMD of those same datasets were divergent, with the SmartMap analyses having considerably higher median HMD across bins of low mappability than the uniread analyses (Figs 5B and S9B).

The difference between mean and median HMD may be attributable to the fact that HMD is a scaled-version of fold-change of IP over Input. We attribute the median HMD divergence to sparser distribution of read depth in the uniread dataset at lower mappability scores (Fig 4B). As such, there are fewer regions with nonzero read depth in both the IP and Input. The result of this mismatch in read distribution is that more regions have an apparent HMD of zero under uniread analysis. That the mean HMDs are similar between the two analyses suggests that the ratios of the total read depths in IP over Input are similar between SmartMap and uniread analyses. Together, these data suggest that the SmartMap analyses preserve the overall HMD across a wide range of mappability scores while also enabling measurement of HMD at a broader range of loci than do uniread analyses.

One of the major benefits of using ICeChIP-seq data is the ability to measure antibody specificity [29–31]. In ICeChIP, internal standards bearing a variety of different histone modifications can be simultaneously spiked into an experiment, and the relative pulldown efficiency of each modification can be quantified as a proportion of the target to measure the off-target binding of the antibody. We wished to determine whether the SmartMap analyses would yield similar specificity estimates as did the uniread analyses. First, we found that the ratio of the reads from the on-target nucleosome in the IP over the Input was highly similar between the uniread and SmartMap analyses (Table 4). Moreover, the scatterplots of specificity (Figs 5C and S10A) and logarithm of specificity (Figs 5D and S10B) under each modality show slopes close to unity and high coefficients of determination. This further shows that the specificity measurements in SmartMap and uniread analyses are highly similar in both an absolute (Figs 5C and S10A) and a relative (Figs 5D and S10B) sense.

## Extending the utility of SmartMap to ATAC-seq and RNA-seq

We also found that SmartMap could be applied to ATAC-seq data to obtain more global measurements of chromatin accessibility. To demonstrate this, we used two replicates of K562 ATAC-seq data, originally generated by the Snyder Lab as part of the ENCODE Consortium [15]. As with the ICeChIP-seq datasets, we found that SmartMap analysis could utilize 17–20%

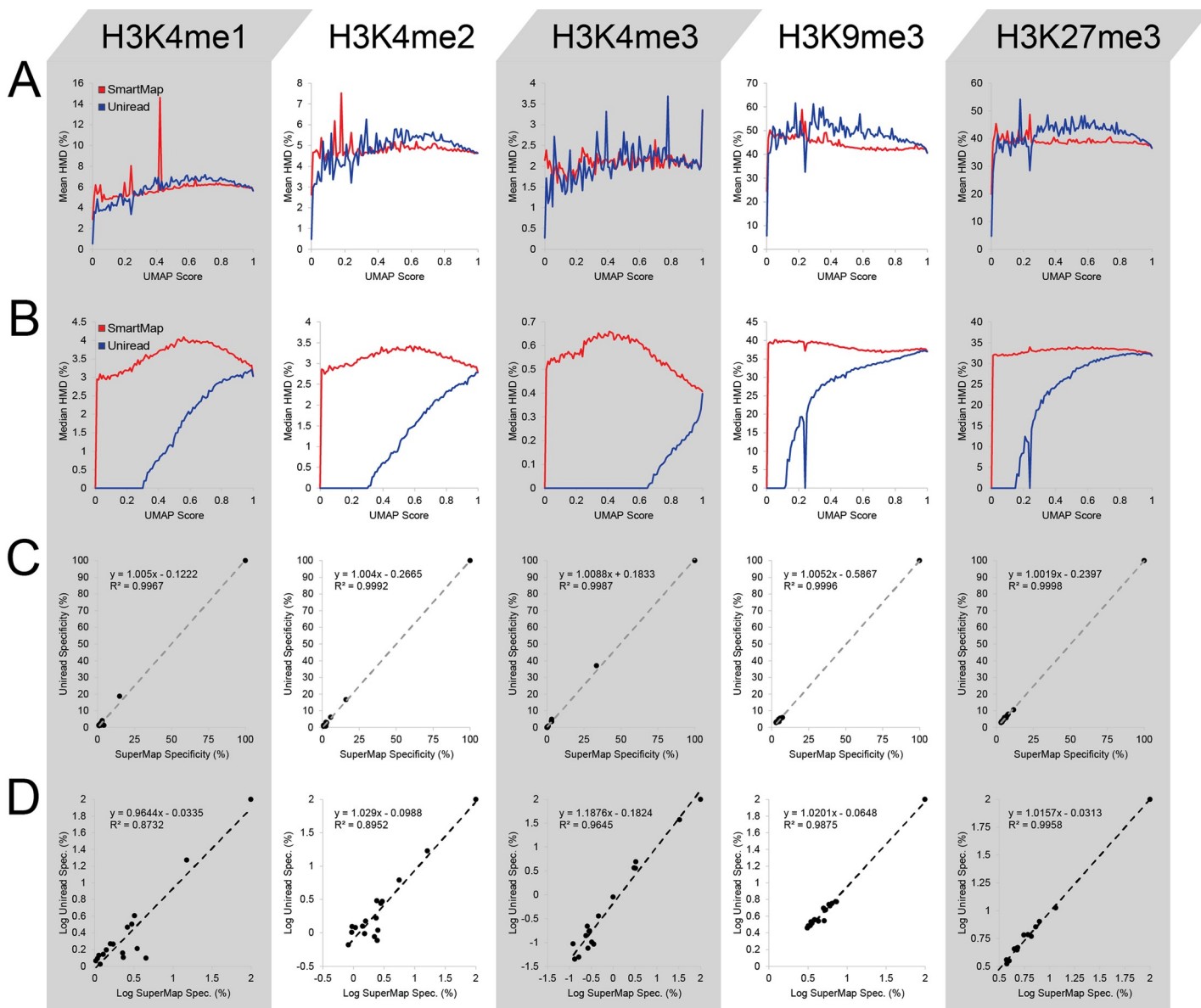

**Fig 5. ICeChIP-seq histone modification density in SmartMap and uniread analyses.** All analyses conducted on 200bp tiled genomic windows. **(A-B)** (A) Mean or (B) Median HMD vs. mappability score (UMAP50) for SmartMap and uniread analyses. **(C-D)** Scatterplots of (C) specificity or (D) log specificity for uniread vs. SmartMap analyses. Specificity is measured as the enrichment of each on- or off-target internal standard nucleosome as a percentage of on-target enrichment.

more reads than uniread analysis (Table 1); this increased read depth was primarily concentrated at 20–30% of the genome (S11A–S11C Fig and Table 2), particularly those loci with low mappability scores (S11D–S11F Fig). SmartMap and uniread analyses also showed similar levels of reproducibility between the two isogenic replicates, though SmartMap showed slightly lower reproducibility between the two datasets than did the uniread analysis (S11G and S11H Fig). These data suggest that SmartMap is also useful for ATAC-seq data and can reveal accessible regions of the genome at poorly mappable loci that would have been missed by uniread analysis alone.

In addition to the MNase-seq, ChIP-seq, and ATAC-seq datasets, we also sought to apply our SmartMap analysis to RNA-seq experiments. Specifically, we analyzed two replicates of

**Table 4. Analysis of ICeChIP Calibrant Barcodes.**

| | Sample | Series | Barcodes | On-target IP/Input Ratio: | | Species | Specificity Plot: | |
|---|---|---|---|---|---|---|---|---|
| | | | | Uniread | SmartMap | | Slope | $R^2$ |
| AR7 | H3K4me3 Rep. 1 | Ser. 1 | 11 | 19.88 | 20.05 | 1 | – | – |
| | H3K4me3 Rep. 2 | Ser. 1 | 11 | 18.95 | 18.99 | 1 | – | – |
| AR8 | H3K27me3 | Ser. 2 | 100 | 0.877 | 0.879 | 1 | – | – |
| AR9 | H3K4me3 | Ser. 2 | 100 | 27.7 | 28.3 | 7 | 1.051 | 0.9984 |
| | H3K9me3 | Ser. 2 | 100 | 1.34 | 1.26 | 7 | 1.012 | 0.9972 |
| | H3K27me3 | Ser. 2 | 100 | 0.678 | 0.677 | 7 | 1.022 | 0.9995 |
| AR16 | H3K4me1 | Ser. 3 | 136 | 4.34 | 4.84 | 17 | 1.005 | 0.9967 |
| | H3K4me2 | Ser. 3 | 136 | 3.98 | 3.75 | 17 | 1.004 | 0.9992 |
| | H3K4me3 | Ser. 3 | 136 | 32.4 | 31.1 | 17 | 1.009 | 0.9987 |
| AR17 | H3K9me3 | Ser. 3 | 136 | 2.45 | 2.23 | 17 | 1.005 | 0.9996 |
| | H3K27me3 | Ser. 3 | 136 | 1.82 | 1.73 | 17 | 1.002 | 0.9998 |

Barcodes: the number of unique DNA barcode sequences in the ICeChIP calibrant series.

Species: the number of distinct modified nucleosomes marked by the barcodes, including the target modification and, if there is more than one species, the off-target modifications.

Specificity plot: summary of the specificity plots shown in Figs 5C and S9A.

K562 bulk RNA-seq data, originally generated by the Gingeras Lab as part of the ENCODE Consortium [15]. Our SmartMap RNA-seq analyses showed that for each replicate, relative to uniread analysis, there was a 20–35% increase in usable reads (Table 1) concentrated into a minority of distinct Refseq genes (S12A–S12C Fig and Table 2). The reproducibility of the two datasets was also similar between the SmartMap and uniread analyses, though as with the ATAC-seq data, the SmartMap analysis showed marginally lower reproducibility between the two RNA-seq experiments than did the uniread analysis (S12D and S12E Fig). With that said, these differences in read depth are relatively minor in magnitude, especially when normalized to differences in read depth in the SmartMap and uniread analyses. Given the other concerns with using this particular multiread allocation algorithm in gapped reads or spliced transcripts (as noted in the Discussion), it is likely that SmartMap is not optimally configured for use in RNA-seq analysis.

## SmartMap drives new biological insights about repetitive DNA elements

With this method, we sought to better explore the role of histone modifications at repetitive regions. Traditionally, the epigenetic profile of repetitive elements is viewed in light of the "genome defense" hypothesis, which suggests that regulation of repetitive elements (and particularly transposable elements) serves to silence the elements and thereby prevent transposition [43]. Consequently, much previous work on this topic has primarily pointed towards repetitive elements being enriched with heterochromatin-associated modifications such as H3K9me2 [44], H3K9me3 [9,43,45–47], and H3K27me3 [43,45,48]. In recent years, some studies have described a role for canonically activating histone modifications at a subset of repetitive elements [49–54]. Indeed, this body of work has suggested that some long interspersed nuclear elements (LINEs) can bear marks such as the transcriptionally activating H3K4me3 modification, particularly early in development [51–53]. Similarly, other work has suggested that a class of mammalian-wide interspersed repeats (MIRs) may be transcriptionally active and play a role in enhancer regulation [54]. Much of this work, however, has relied upon uncalibrated ChIP with antibodies of uncertain specificity, both of which can result in

data distortion and biologically incorrect conclusions [30]. Further, the ChIP-seq and RNA-seq studies have used a variety of different methods of aligning and filtering for reads to reach their conclusions, none of which used a method similar to our Bayesian SmartMap analysis, which may further affect the interpretations of the experiments. As such, we sought to use our calibrated and highly specific ICeChIP-seq datasets in conjunction with SmartMap to gain new insights into the epigenetic landscape of repetitive elements and to examine the degree to which uniread analysis yields an incomplete view of the data.

To accomplish this, we examined the histone modification landscape at the promoters of all repetitive elements, LINEs, short interspersed nuclear elements (SINEs), and Simple Repeats. K-means clustering analysis on all repetitive elements revealed four classes of promoters, each with a different histone modification profile: Cluster 1, enriched for H3K27me3 and H3K9me3; Cluster 2, enriched for H3K4me1 and H3K4me2; Cluster 3, enriched for H3K4me2 and H3K4me3; and Cluster 4, which is relatively depleted of histone modifications (Fig 6A). These clusters are roughly reminiscent of the functional classifications of the ENCODE hidden Markov model, where Clusters 1, 2, and 3 correspond to silenced promoters, enhancers, and active promoters, respectively [55]. Interestingly, in all but Cluster 4, a greater proportion of nucleosomes is enriched with H3K27me3 than H3K9me3, despite the previous emphasis on the latter in repetitive element silencing [9,43,45–47], emphasizing the importance of calibration in ChIP-seq studies for comparing different modifications [29,30]. Similar histone modification profiles are seen for the LINEs (S13A Fig), SINEs (S13B Fig), and Simple Repeats (S13C Fig). Across all these classes, Cluster 3 had the highest ATAC-seq signal (S13D–S13G Fig), consistent with the presence of histone modifications associated with transcription and accessible chromatin [10,13,56].

Importantly, SmartMap analysis enabled us to more accurately measure HMD and assign clusters than did unread analysis. Overall, there were 142,392 promoters with nonzero HMD in the SmartMap analysis that displayed no measurable HMD within 200bp of the TSS across all five histone modifications in the unread dataset; similarly little HMD was detected within 1kb of the same in the unread dataset (S14 Fig). This increase in HMD was substantial; most such sites had meaningful levels of histone modifications (Fig 6B). A small subset was primarily H3K4me2/me3 predominant; a larger subset had high levels of H3K4me1/me2, and the remainder were primarily characterized by H3K27me3 and H3K9me3 (Fig 6B). These represent promoters that would have been misclassified as histone-modification-depleted under unread analysis; it is only through proper allocation of multireads that we can measure their HMDs and assign them to the appropriate cluster of repeat elements.

The distribution of repetitive elements across these clusters revealed interesting patterns. The distribution of the repeat classes or families across the clusters are presented in Table 5 for all repeats, Table 6 for LINEs, and Table 7 for SINEs, and summarized in Fig 6C. Notably, amongst SINEs, MIRs were enriched in Cluster 3 (Fig 6C), consistent with previous descriptions of a class of transcriptionally active MIRs [54]. In addition, Cluster 3 was enriched for Simple Repeats across all repeat promoters, consistent with descriptions of Simple Repeats in and around protein coding genes in the literature [57]. Interestingly, Cluster 3 was enriched for the L2 subtype of LINEs, despite previous work primarily focusing on the role of H3K4me3 at L1 elements [52], representing a novel prediction of transcriptional activity of this family. To this end, using SmartMap analysis of the RNA-seq data, we found that the Cluster 3 LINEs had greater transcriptional activity than did the other clusters (Fig 6D), confirming the transcriptional activity suggested by the presence of H3K4me3. Collectively, these data demonstrate the risk in only focusing on unreads–namely, the risk of missing important classes of genomic features–and highlights the role of multiread analysis of both DNA and RNA in driving new biological discovery.

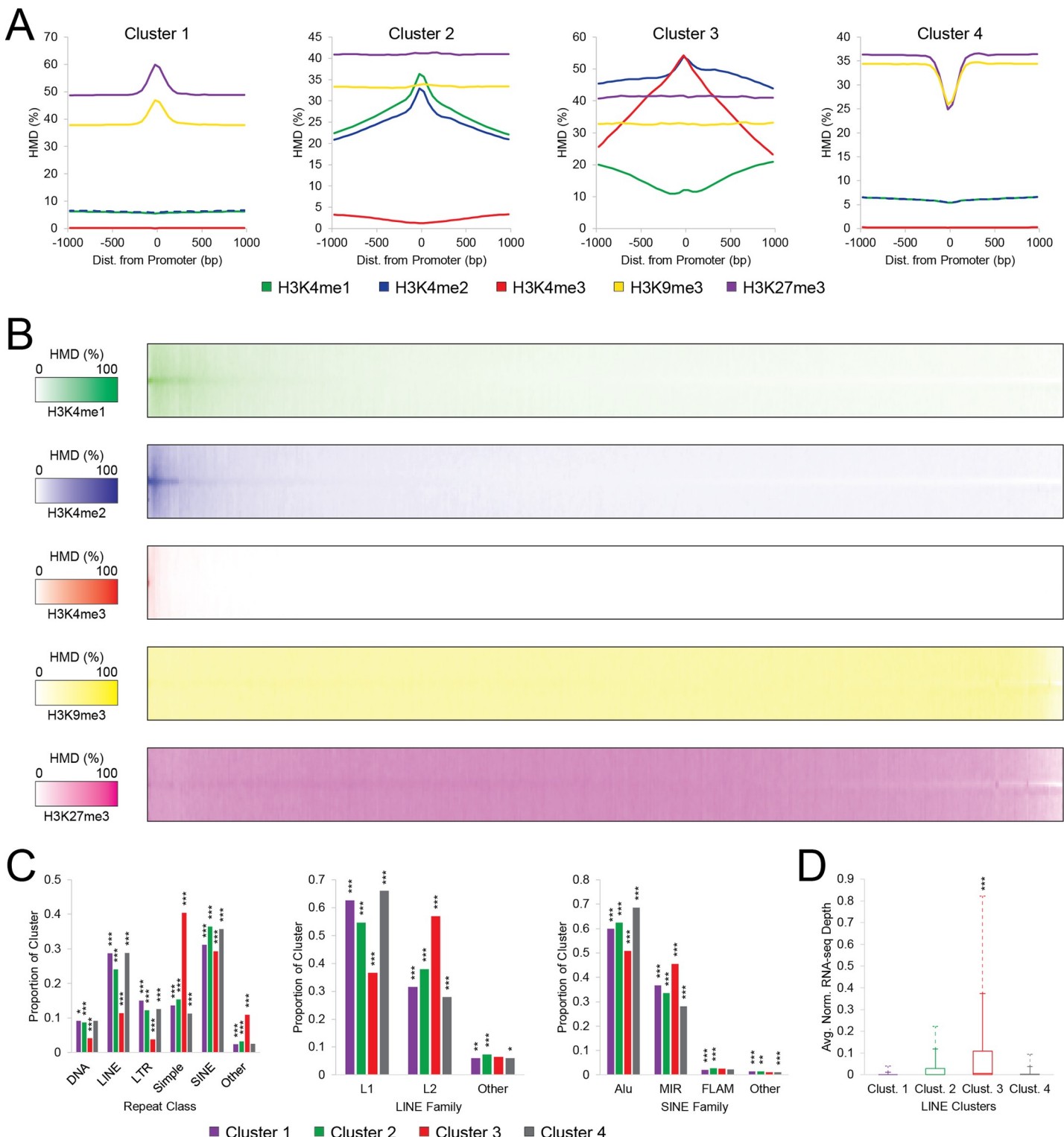

**Fig 6. Assessment of histone modifications at promoters of repetitive DNA elements. (A)** Mean histone modification densities (HMDs) about promoters for classes of all repetitive elements, as defined by k-means clustering. Corresponding analyses of LINE, SINE, and Simple Repeat elements in S13 Fig. **(B)** Heatmap of repeat promoters with newly measurable HMD in SmartMap analysis, sorted on first principal component of repetitive elements. **(C)** Proportion of each cluster comprised by each repeat class or family for all repeats (left), LINE elements (center), and SINE elements (right). All significance tests performed as post-hoc Bonferroni-corrected pairwise 2x2 chi-square tests. **(D)** Quantile boxplots of average normalized RNA-seq read depth across LINE elements for each LINE cluster. Solid line with marker represents 90th percentile; dashed line with marker represents 95th percentile. Significance test shows difference in median by Bonferroni-corrected pairwise Mood's median tests. Significance markers: $^{*}p<0.01$, $^{**}p<10^{-5}$, $^{***}p<10^{-10}$.

**Table 5. Clustering of Repetitive Elements.**

| Repeat Class | Cluster 1 | Cluster 2 | Cluster 3 | Cluster 4 | Total |
|---|---|---|---|---|---|
| DNA | 240,787 | 28,465 | 763 | 232,324 | **502,339** |
| LINE | 754,991 | 78,670 | 2,133 | 734,556 | **1,570,350** |
| LTR | 393,797 | 39,911 | 707 | 319,769 | **754,184** |
| Simple Repeat | 357,734 | 50,395 | 7,555 | 287,900 | **703,584** |
| SINE | 818,624 | 119,404 | 5,478 | 908,873 | **1,852,379** |
| Other | 61,336 | 10,461 | 2,038 | 62,635 | **136,470** |
| **Total** | **2,627,269** | **327,306** | **18,674** | **2,546,057** | **5,519,306** |

## Discussion

In this work, we have described a method to markedly increase sequencing depth genome-wide by analyzing ambiguously mapped reads rather than discarding them. This is of particular importance given that a significant portion of commonly studied genomes are not uniquely mappable by single-end or paired-end sequencing [25,26]. This difficulty arises in no small part due to the repetitiveness of the genome [22], but despite their difficulty to map, repetitive elements play critical roles in genomic regulation and function [27]. It is common discard these multireads entirely, despite these reads representing up to 30% of the sequencing depth. Works that do utilize multireads often simply select an alignment at random. We demonstrate that our SmartMap algorithm can better map reads to the repetitive portion of the genome, facilitating better understanding their functions. Importantly, we find that the usage of alignment quality scores and paired-end sequencing can markedly increase the accuracy of imputed alignments.

Just by incorporating multireads with 2–50 alignments, we were able to increase the read depth of our samples by 13–53% (Fig 7A and Table 1). This increase in read depth was not simply distributed across the entire genome, which is critical for the usefulness of this method. If the multireads were distributed uniformly, it would only modestly decrease error by slightly increasing read depth at all loci [29]. However, that is not the case; rather, the multireads are concentrated in a minority of the genome (Fig 7B and Table 2), bringing regions of lower mappability to read depths comparable with highly mappable loci (Fig 4A). The multiread samples have a 5–20% increase over unireads in the number of loci with nonzero read depth (Fig 7C and Table 2), representing a sizable proportion of the genome that is completely ignored by uniread analysis and can be recovered only by utilizing ambiguously mapped reads.

Our method requires no particular experimental modifications or additional controls for analysis of multireads and can be applied post hoc to existing datasets. As such, SmartMap can be used to leverage the existing compendium of NGS datasets more accurately. Though we primarily used ICeChIP-seq data to demonstrate and explore the capabilities of SmartMap, this tool is not solely designed for ICeChIP and does not require the internal standards used therein. Indeed, SmartMap is designed to be a general tool for a broad range of next-generation sequencing experiments, including ChIP-seq, MNase-seq, and ATAC-seq, as we showed

**Table 6. Clustering of LINE Elements.**

| LINE Family | Cluster 1 | Cluster 2 | Cluster 3 | Cluster 4 | Total |
|---|---|---|---|---|---|
| L1 | 463,275 | 46,376 | 828 | 490,788 | **1,001,267** |
| L2 | 233,599 | 32,236 | 1,289 | 207,410 | **474,534** |
| Other | 43,993 | 6,221 | 146 | 44,189 | **94,549** |
| **Total** | **740,867** | **84,833** | **2,263** | **742,387** | **1,570,350** |

**Table 7. Clustering of SINE Elements.**

| SINE Family | Cluster 1 | Cluster 2 | Cluster 3 | Cluster 4 | Total |
|---|---|---|---|---|---|
| Alu | 520,927 | 72,223 | 2,880 | 590,748 | **520,927** |
| MIR | 319,283 | 38,836 | 2,587 | 241,855 | **319,283** |
| FLAM | 17,988 | 3,069 | 148 | 18,782 | **17,988** |
| Other | 11,943 | 1,634 | 61 | 9,415 | **11,943** |
| **Total** | **870,141** | **115,762** | **5,676** | **860,800** | **870,141** |

in this work. In addition, though we have used paired-end sequencing here, there is little reason to believe this method could not be used for a single-end sequencing experiment. In principle, an algorithm using the principles of SmartMap can be applied to any NGS experiment, past or future, that involves alignment to a genome.

Previously published methods have utilized a variety of techniques to allocate multireads; however, our analysis suggests that many of these methods may be problematic. One heuristic assumes that multireads and unireads have similar genomic distributions and, accordingly, assigned multireads weights in proportion with uniread depth [2,58]. Our data, by contrast, shows that multireads instead concentrate into a minority of loci (Table 2) and particularly those with low uniread depth (Figs 3C and S2C and S2D). This suggests that the unireads and multireads have different genomic distributions, violating the critical assumption underlying proportional allocation of multireads. Another method of resolving multireads is to select one alignment at random for each read [39,43]. The expected value of the read distribution under this procedure converges to that of SmartMap without reweighting, which we found to have higher error than SmartMap with a Bayesian reweighting cycle (Figs 2D and 3H and Table 8). Indeed, explicit comparison to an instance of random read comparison revealed even higher error as compared to both un-reweighted SmartMap and the SmartMap analysis with reweighting (Table 8).

SmartMap is also computationally efficient as compared to the most similar previous algorithms and software for the assignment of multireads. This is due in part to the low number of reweighting iterations our algorithm uses, which decreases the computational burden of the software. In addition, the Fenwick tree data structure used with our method allows for more efficient processing of reads by accessing and updating of genomic weights. Previous implementations of a scored-alignment reweighting algorithm, as done by BM-Map, have required more than five hours to process approximately seven million aligned reads after alignment in previous studies [36]. Unfortunately, we were not able to fully measure the time requirements for BM-Map for ourselves; implementing both CSEM [35] and BM-Map [36] proved challenging, as described in the Methods. However, using our simulated dataset (with 50bp reads), including more than 740 million alignments from more than 275 million reads, even just reading the alignments with BM-Map on our hardware took more than 6 hours after alignment (Table 8). By contrast, our algorithm can completely process that same aligned dataset in less than 2.5 hours, representing more than 100-fold less CPU time than the alignment itself (Table 8). As such, the low CPU-time requirements of SmartMap drastically increases our ability to use this algorithm on data from modern NGS experiments, particularly given the ever-increasing depth and decreasing cost of sequencing [59]. Though it is, admittedly, faster to solely process unireads than to conduct SmartMap (Table 8), the added time is not egregiously high; on our system, the full benchmarking (including alignment) required roughly eight more hours of wall time in the SmartMap analysis than in the uniread analysis.

This SmartMap method is, however, not without its limitations. Primary amongst these limitations is that rather than yielding a list of alignments, the SmartMap software either

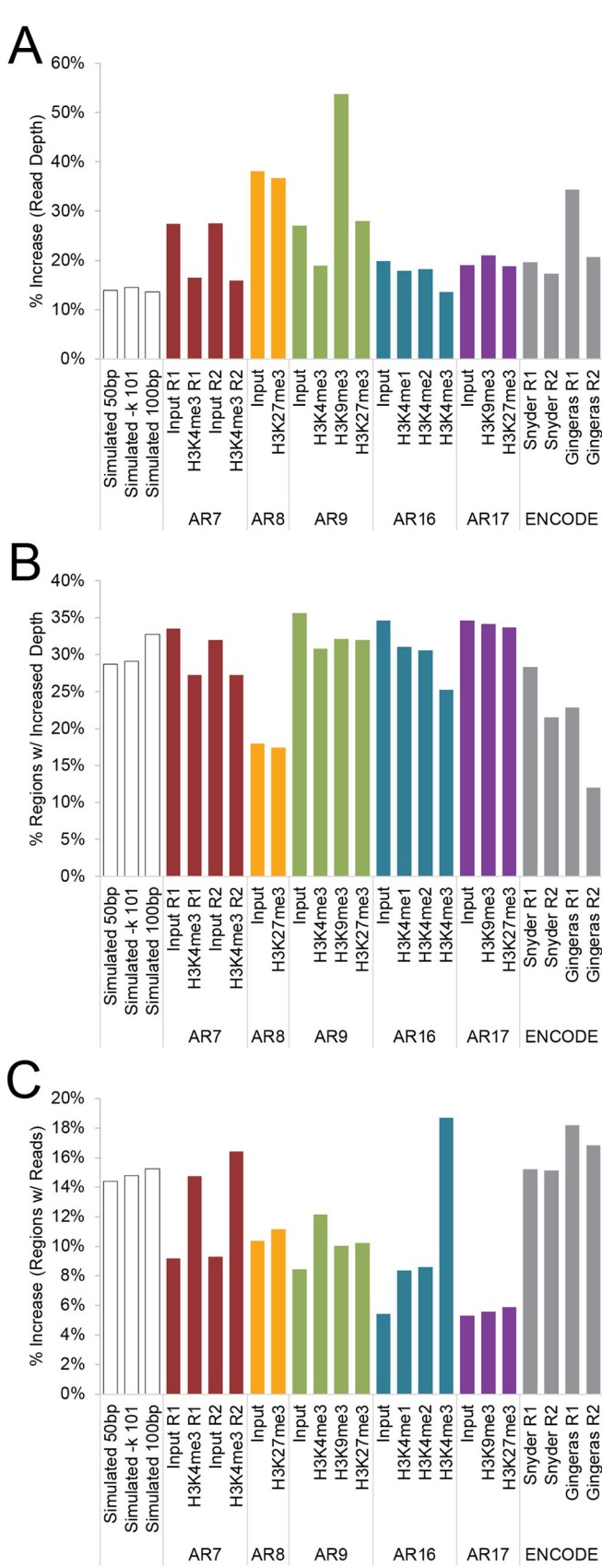

**Fig 7. Analysis of increased usable read depth.** This figure graphically represents the data in Tables 1 and 2. **(A)** The percent increase in the number of reads usable in SmartMap analysis (reads with 1–50 alignments) relative to uniread analysis (reads with 1 alignment). **(B)** Percentage of the total number of regions with an increase in read depth in the SmartMap dataset relative to the uniread dataset. For all datasets except the ENCODE RNA-seq datasets, the list of regions analyzed is the set of 200bp genomic windows across the relevant genome (hg38, mm10, or dm3). For the ENCODE RNA-seq dataset, the list of regions analyzed is the set of distinct Refseq genes. **(C)** Percent increase in the number of regions with nonzero read depth in the SmartMap dataset relative to the uniread dataset. Regions are defined as per panel B.

outputs the read depth at each base pair genome-wide or a list of alignments with associated weights. While this is useful for any analysis that utilizes the read depth at a given position, this makes it difficult to use downstream methods or tools that primarily utilize the full list of alignments using off-the-shelf tools. In particular, this makes it challenging to compute gene expression in RNA-seq per common methods such as FPKM, which uses the number of reads overlapping a transcript as a measure of expression rather than the read depth per base pair. This is partially alleviated by the fact that SmartMap provides the option to write lists of alignments with their corresponding weights, but even so, incorporating these weights into existing downstream analyses, pipelines, and software may remain challenging.

In addition, the SmartMap method also may face challenges with any alignments with significant gaps relative to the alignment templates, such as those created by RNA splicing (or Hi-C experiments). Because our reweighting algorithm assigns weights based on the average read depth across an alignment, an alignment spanning a splice junction in RNA-seq may be unfairly assigned a lower weight due to decreased read depth in the intron. As such, highly spliced genes may be given a lower read depth than a similarly expressed gene with fewer introns. This could be partially accommodated by weighting with the total read depth over an alignment rather than the average read depth over the same, but this method would potentially unfairly increase weight of longer alignments, which could pose another challenge.

In addition, from a computational perspective, the SmartMap method is memory intensive. This is in large part due to the data structure used for storing genome-wide weight data. Because this tool is designed to be compatible with reweighting of paired-end reads and obtaining the total weight across a paired-end read, the data structure needs to efficiently conduct both range update and range query operations. Accordingly, for the strand-independent method, we have used a dual binary-indexed tree data structure; for strand-specific analysis,

**Table 8. Benchmarking SmartMap software.**

| | Pre-algorithm alignment and processing | | | | Read allocation algorithm | | | |
| | Read Alignment | | Processing Alignments | | Reading alignments | | | |
| Algorithm | CPU Time | Wall Time | CPU Time | Proc. File Size | CPU Time | Max. Memory | Algorithm Time | Avg. MAE |
|---|---|---|---|---|---|---|---|---|
| SmartMap | 317:30:25 | 6:39:46 | 1:34:29 | 59 GB | 0:16:49 | 53 GB | 0:42:38 | 4.04 |
| BM-Map | 317:30:25 | 6:39:46 | N/A | 820 GB | 6:25:09 | 146 GB | ERROR | – |
| Iteration 0 | 317:30:25 | 6:39:46 | 1:34:29 | 59 GB | 0:16:10 | 53 GB | 0:35:16 | 4.12 |
| Random | 317:30:25 | 6:39:46 | 2:15:12 | 15 GB | 0:03:58 | 39 GB | 0:15:46 | 5.48 |
| Uniread | 36:08:04 | 0:45:34 | 0:17:07 | 13 GB | 0:03:19 | 39 GB | 0:14:43 | 6.50 |

Benchmarking conducted on computer with Ubuntu 20.04.1 LTS with 224 GB of RAM and dual Intel Xeon CPU E5-2690 v3 @ 2.60GHz processors, running on one thread except the read alignment, which used 48 threads. All times are represented in hours:minutes:seconds.

Alignment conditions are identical for all but Uniread. Parsing reads is typically conducted in parallel with alignment. File size represents the size of the required file after read parsing needed for the algorithm in question. Reading alignments is part of each algorithm and is included in the Algorithm Total Time.

Average Mean Absolute Error (Avg. MAE) is computed against the gold standard on the set of true origin loci. These benchmarking analyses were conducted separately with separate alignments from the analyses in Fig 3, and the avg. MAEs vary slightly in magnitude from those presented in Fig 3.

we use a dual binary-indexed tree structure for each strand, for a total of four binary-indexed trees. For this reason, for our simulated dataset, the SmartMap analysis required almost 60 GB of memory. In principle, a lower-memory method could be developed that would only use one binary-indexed tree per strand, but this would require iteration over each base position of each alignment and would thereby dramatically decrease time-efficiency. However, it's important to note that BM-Map, the only other tested software that was even able to successfully read the alignments, required almost 150 GB to read that same set of alignments into memory (Table 8). In practice, with the decreasing costs of memory and the increasing availability of computational servers and clusters for a wide variety of bioinformatic tools and analyses, the memory requirements are likely workable for many users, particularly because of the low CPU time required.

Finally, even the best SmartMap analysis can only be as good as the alignment itself. In this work, we have largely restricted our Bowtie2 alignments to report a maximum of 51 alignments, with the exception of the analysis with a maximum of 101 alignments. This was conducted for feasibility; as the maximum number of reported alignments increases, so too does the computational overhead needed for alignment of the reads by Bowtie2. However, this does place an inherent limitation on our ability to look at the most repetitive regions of the genome, which can be found at hundreds of loci throughout the genome and can thus pose a significant challenge to alignment and multiread analysis. Granted, raising this threshold to a maximum of 101 alignments per reads had practically no impact on the analysis on the human genome (S5 Fig and Tables 1 and 2), but nonetheless, there were still nearly seven million reads that aligned to the maximum of 101 loci, representing a significant number of reads with even more potential alignments. Further, some genomes have even greater repetitiveness than does the human genome; for example, repetitive elements comprise roughly 85% of the maize genome [60], making alignment all the more challenging and raising the number of plausible alignment sites for each read. It is important to note that this is not a limitation that is inherent to SmartMap, but rather, to alignment itself. If the end user was able to generate an alignment with an arbitrarily high maximum number of reportable alignments, there is no reason that SmartMap should fail; it is not inherently capped at a maximum number of alignments per read. It should be remembered that SmartMap will not be able to "fix" an alignment with too few alignments per read. Accordingly, it may be necessary to tune the maximum number of alignments per read to appropriately analyze data originating from some genomes despite the added computational load for alignment.

Despite these limitations, we were nonetheless able to demonstrate the usefulness of our SmartMap tool to process reads from a variety of NGS workflows (e.g. MNase-seq, ChIP-seq, ATAC-seq, and RNA-seq) and to investigate biological questions–in this case, the epigenetic regulation of repetitive elements. Just as importantly, we demonstrated the risk of using only unireads–namely, that biologically relevant regions will be hidden from analysis because the multireads have been discarded. Given the critical role that repetitive regions play in biological regulation [27], being able to analyze these regions is crucial to gaining a more complete understanding of genomic structure and function. Accordingly, we hope this method will help enable researchers to use their sequencing data more completely and thereby gain more useful information from their experiments.

## Methods

### Sequencing data sources

**MNase-seq and ICeChIP-seq data.** The ICeChIP-seq datasets analyzed in this work, with the exception of AR17 H3K27me3 IP, were sourced from previously published ICeChIP-seq datasets [29,30]. The FASTQ files for datasets AR7, AR8, and AR9 can be obtained from GEO

Accession Number GSE60378. The FASTQ files for datasets AR16 and AR17 can be obtained from GEO Accession Number GSE103543. The Inputs for each ICeChIP experiment are generated by MNase-seq.

The AR17 H3K27me3 ICeChIP-seq was conducted at the same time as the AR17 H3K9me3 ICeChIP-seq experiment using the same AR17 Input, but was not published previously [30]. It was generated by ICeChIP-seq as previously described [30] using an anti-H3K27me3 antibody (Cell Signaling Technologies, Product Number 9733, Lot 8). This dataset has been made available as a newly added series at GSE103543.

**RNA-seq data.** RNA-seq data was obtained from the ENCODE Project [61] from experiment ENCSR000AEL. The FASTQ files for Isogenic Replicate 1 was obtained from the dataset for library ENCLB053ZZZ (FASTQ accession numbers: ENCFF001RFF, ENCFF001RFE). The FASTQ files for Isogenic Replicate 2 was obtained from the dataset for library ENCLB054ZZZ (FASTQ accession numbers: ENCFF001RFD, ENCFF001RFC).

**ATAC-seq data.** ATAC-seq data was obtained from the ENCODE Project [61] from experiment ENCSR483RKN. The FASTQ files for Isogenic Replicate 1 was obtained from the dataset for library ENCLB918NXF (FASTQ accession numbers: ENCFF391BFJ, ENCFF186CQZ). The FASTQ files for Isogenic Replicate 2 was obtained from the dataset for library ENCLB758GEG (FASTQ accession numbers: ENCFF440UAD, ENCFF350ZZR).

**Mappability scores.** The mappability score chosen was the UMAP50 score, which represents the proportion of 50bp kmers overlying a given point that are unique in the genome [25]. The approximate UMAP50 score of the dm3 genome was computed by computing all 50-mers in the genome and determining those that are unique; the genome coverage of the unique 50-mers was then determined to compute approximate UMAP50 score of the genome.

**Simulated dataset.** The simulated dataset was generated as followed. First, 6 million loci of length 200bp in the genome were randomly selected and designated as the target loci. Paired-end Illumina sequencing reads were then simulated using NEAT-genReads [62] using the list of target loci as the target file and the following settings: 50bp read length, 30-fold target coverage, default off-target coverage, and insert size 175bp average and 10bp standard deviation. The output list of true read locations was then used to compute a Gold Standard genome coverage Bed-Graph using BEDTools genomecov [63]. The average Gold Standard read depth of the target loci was then computed as described below, and the target loci with nonzero Gold Standard read depth were designated as the "true origin" loci and used for downstream analysis.

To generate the simulated dataset with 100bp read length, the same procedure was used on the same set of 6 million loci, with the NEAT-genReads tool being set to output 100bp reads rather than 50bp reads. Unless otherwise specified, this work uses "simulated dataset" or similar to refer to the simulated dataset with 50bp reads.

## Computing average value of BEDGRAPH at target loci

Because the BEDTools map software does not compute base-pair-wise averages of BED-GRAPH signals, the following procedure was used to compute read depth at a list of target loci. Overlapping loci were merged using BEDTools merge, and the resultant list of loci were partitioned into 1bp windows using BEDTools makewindows. The BEDGRAPH was then mapped onto the windows using BEDTools map, and the mapped windows were then mapped with the mean function onto the original list of target loci using BEDTools map.

## Mappability estimation and binning

The mappability of a list of loci was computed by computing the average value of the UMAP50 bedgraph for the relevant genome at those loci using the method described above. To compute

the number of regions per UMAP50 score, the loci were binned by average UMAP50 score in bins of width 0.01. The number of loci at each bin were then computed to determine the approximate distribution of UMAP50 score across the selected loci.

## MACS2 peak calling

Peak calling was conducted on the simulated datasets using MACS2 [64] with the bdgpeakcall function with the relevant BEDGRAPH file and default settings.

## Alignment and read filtering and processing

FASTQ files for ICeChIP-seq or the simulated dataset were aligned using Bowtie2 [40] due to its common usage in the field and due to its ability to report alignment scores for each mate for each alignment reported as opposed to for just the best alignment. Bowtie2 alignment was run on the paired-end sequencing samples with the following settings: end-to-end alignment, very-fast preset settings, no discordant alignments, no mixed alignments, report up to 51 alignments, insert size minimum 100bp, insert size maximum 250bp. In the case of the analysis with up to 101 alignments (the k101 dataset), the above settings were used with up to 101 alignments reported per read. The genomes used for alignment were as follows: AR7, mm10 with ICeChIP barcodes series 1; AR8, dm3 with ICeChIP barcodes series 1; AR9, mm10 with ICeChIP barcodes series 2; AR16 and AR17, hg38 with ICeChIP barcodes series 3; simulated datasets, hg38.

FASTQ files for RNA-seq were aligned using Hisat2 [65] for the same reasons as the choice to use Bowtie2. Hisat2 alignment was run on the paired-end sequencing samples with the following settings: no discordant alignments, no mixed alignments, report up to 51 alignments. The genome used for alignment was hg38 with the ENCODE ERCC standards.

FASTQ files for ATAC-seq were aligned using Bowtie2 [40] on the paired-end sequencing samples with the following settings: no discordant alignments, no mixed alignments, report up to 51 alignments, insert size maximum 2000bp.

Alignments were then filtered to select for reads that are paired, mapped in a proper pair, and mate on the reverse strand, corresponding to SAM flags of 99, 163, 355, and 419. For non-strand-specific applications, the selected SAM file records were then extracted into a file containing the following fields: chromosome, start position, stop position, read ID, read alignment score (field labeled "AS:i:"), mate alignment score (field labeled "YS:i:"). For strand-specific applications, the selected SAM file records were extracted into a file containing the following fields: chromosome, start position, stop position, read ID, strand, read alignment score (field labeled "AS:i:"), mate alignment score (field labeled "YS:i:"). The reads were then split into separate BED files based on the number of alignments per read. For downstream uniread analyses, only the reads with 1 alignment were used; for downstream SmartMap analyses, reads with 1–50 alignments were used except for the k101 dataset, in which case reads with 1–100 alignments were used.

The file with 51 alignments per read (or that with 101 alignments per read for the k101 analysis) was not used for downstream analyses to prevent confounding by reads with fewer reported than possible reads.

## Uniread and SmartMap analysis of genome coverage

For the uniread analysis, our SmartMap software was used with only the file containing reads with only 1 alignment per read. For the SmartMap analysis, our SmartMap software was used with the files containing reads with fewer than 51 alignments per read.

The SmartMap software uses a set of dual Binary Indexed Trees to store map counts and weights across the genome and uses an iterative Bayesian reweighting algorithm to assign weights to each of the different alignments. These steps are outlined below. Unless otherwise specified, all analyses are conducted with 1 iteration in scored mode. For the strand-specific analyses, there is a separate set of dual Binary Indexed Trees for each strand.

**Storage of map counts in the genome.** To facilitate efficient summation and updating of map counts and weights across the genome, each chromosome is stored as a pair of Binary Indexed Trees (BIT), also known as Fenwick Trees. The BIT is a data structure that is efficient for computing prefix sums of an ordered dataset from the beginning of the dataset to the given index. Because we used a 1-based coordinate system for the genome, the datasets to which we refer as being represented by a BIT should be assumed to be 1-based datasets unless otherwise specified.

For a dataset of length $L$, the BIT is represented by $L+1$ nodes, which are stored in an array. To increment a dataset represented by a BIT $T$ at index $i$ by the value $v$, the following algorithm is used. Let $T[i]$ represent the $i$th node of $T$. Let $lsb(i)$ represent the lowest significant bit in the binary representation of $i$. Then:

$$T[i] = T[i] + v \tag{1}$$

$$i = i + lsb(i) \tag{2}$$

If the new value of $i \leq L+1$, Eqs 1 and 2 are iterated until $i > L+1$. For brevity, we will refer to this operation to increment the BIT $T$ representing the 1-based dataset by $v$ in the value $i$ as *BITUpdate(T, i, v)*.

To compute the prefix sum of the dataset at index $i$ (i.e. the sum of all values with indices *[1, i]* of a 1-based dataset), the following algorithm is used, using the above definitions of $T[i]$ and $lsb(i)$. Let the prefix sum be represented by *sum*, where *sum* = 0 at the beginning of the algorithm. Then:

$$sum = sum + T[i] \tag{3}$$

$$i = i - lsb(i) \tag{4}$$

If the new value of $i > 0$, Eqs 3–4 are iterated until $i \leq 0$. For brevity, we will refer to this operation to obtain the prefix sum of the $T$ at value $i$ as *BITSum(T, i)*.

To understand how we here use BITs to efficiently store values across the genome and efficiently sum the values across loci, consider the following.

Consider a dataset $C$ represented by BITs $T_1$ and $T_2$. If the values of $C$ for indices in range *[l, r)* are incremented by $v$, then let the resulting dataset be represented by $C'$. Let *ΔPointSum (C, i) = PointSum(C', i)–PointSum(C, i)*.

The prefix sum of the resulting dataset $C'$ at index $i$, represented by *PointSum(C', i)*, is changed in one of three ways:

**Case 1:** $i < l$. In this case, the increment on range *[l, r)* will not change *PointSum(C', i)*. As such:

$$\Delta PointSum(C, i) = 0 \tag{5}$$

**Case 2:** $l \leq i < r$. In this case:

$$PointSum(C', i) = PointSum(C, l - 1) + v * (i - (l - 1))$$

However, per Eq 5, *ΔPointSum(C, l-1) = 0*. As such:

$$\Delta PointSum(C, i) = v * i - v * (l - 1) \tag{6}$$

**Case 3:** $i \geq r$. In this case, the increment on range *[l, r)* will not change the values of *C* past index *r-1*. Accordingly:

$$PointSum(C', i) = PointSum(C, i) + (PointSum(C', r) - PointSum(C, r))$$

$$\Delta PointSum(C', i) = \Delta PointSum(C, r)$$

However, per Eq 6, $\Delta PointSum(C, r) = v*r - v*(l-1)$. As such

$$\Delta PointSum(C, i) = (v + (-v)) * i - (v * (l - 1) - v * r) \tag{7}$$

These three cases will provide the basis for our use of BITs to store and efficiently sum values across the genome. Each chromosome *C* in the genome is stored as a pair of BITs, to which we shall here refer as $T_1$ and $T_2$. Let *L* represent the length of the chromosome. Accordingly, $T_1$ and $T_2$ have *L+1* nodes.

To increment the value associated with the base pairs in the range *[l, r)* by an increment value *v*, the following procedure is used.

$$BITUpdate(T_1, l, v)$$

$$BITUpdate(T_1, r, -v) \tag{8}$$

$$BITUpdate(T_2, l, v * l)$$

$$BITUpdate(T_2, r, -v * r)$$

The value associated with base pair *i* is then *BITSum(T₁, i)*.

To obtain the prefix sum of the chromosome dataset *C* at base pair index *i*, represented by *PointSum(C, i)*, the following equation is used.

$$PointSum(C, i) = BITSum(T_1, i) * i - BITSum(T_2, i) \tag{9}$$

The sum of the values associated with each base pair in the range *[l, r) = [l, r-1]* on chromosome *C*, represented by *LocusSum(C, l, r)*, can thus be described by:

$$LocusSum(C, l, r) = PointSum(C, r - 1) - PointSum(C, l - 1) \tag{10}$$

This dual-BIT data structure allows for efficient handling of data with respect to time complexity. The *BITUpdate* and *BITSum* steps occur with time complexity O(log L), and the updates to ranges (Eq 8) and range summations (Eq 10) use four *BITUpdates* and four *BITSums*, respectively. As such, both range updates and range queries occur with time complexity O(log L).

**Iterative Bayesian reweighting of mapped reads.** To assess and appropriately weight reads mapped to different portions of the genome, we implemented a Bayesian approach which iteratively reweights the mappings associated with each read. For each read, we assign to each associated map a weight representative of the prior probability that the map is the origin of the associated read. We then iteratively use the distribution of the assigned maps and their weights (the prior probabilities) to determine the posterior probability for each map

being the true origin of the associated read and assign that as the weight for that map, which then becomes the prior probability for the next iteration.

Let the set of all sequencing reads be represented as $R$, with an individual sequencing read being represented as $r_i$. Then $R = \{r_i \ldots r_n\}$, where $n$ represents the total number of sequencing reads obtained for the dataset in question.

Each read $r_i$ is associated with a true genomic origin locus $l_i$ and a set of genomic alignments $M_i = \{m_{i,1} \ldots m_{i,k}\}$, where each $m_{i,j}$ represents a reported alignment of $r_i$ and $k$ represents the total number of alignments reported for $r_i$ such that $k < k_{max}$, the maximum number of possible reported alignments. Each reported alignment $m_{i,j}$ is associated with an alignment score $s_{i,j}$, a weight $w_{i,j}$, and an alignment genomic locus $g_{i,j}$. We will define the set of all alignment scores associated with read $r_i$ as $S_i = \{s_{i,1} \ldots s_{i,k}\}$, with the set of all alignment weights associated with read $r_i$ being represented as $W_i = \{w_{i,1} \ldots w_{i,k}\}$, and with the set of all alignment loci associated with read $r_i$ being defined as $G_i = \{g_{i,1} \ldots g_{i,k}\}$.

For this algorithm, we assume that for each alignment $m_i$ associated with a given read $r_i$, one of the associated alignment loci $g_{i,j}$ is the true origin locus $l_i$. Then each weight $w_{i,j}$ is defined as the probability $w_{i,j} = Pr(g_{i,j} = l_i)$, or the probability that the alignment associated with the weight $w_{i,j}$ is equal to the true origin locus.

The set of all true genomic origin loci $l_i$ will be defined as $L = \{l_1 \ldots l_n\}$. The set of all alignment scores, weights, and loci associated with every read in $R$ will be defined as $S = \{S_1 \ldots S_n\}$, $W = \{W_1 \ldots W_n\}$, and $G = \{G_1 \ldots G_n\}$.

These variables will define our analysis. Our observed variables are the set of alignment scores $S$ and the set of alignment loci $G$. Our latent variable is the true genomic origin distribution $L$. We will be modeling to generate the set of alignment weights $W$ with the goal of estimating the true read origin distribution $L$ as the expected value of the set of reported alignments $G$ with the set of weights $W$ being treated as the probability distribution of $G$ upon which the expected value is computed.

When conducting analyses in "scored mode," we wish to consider the quality of each alignment. To do this, for each alignment $g_{i,j}$ of each read $r_i$, we will transform the associated alignment score $s_{i,j}$ into a pseudo-MAPQ score $z_{i,j}$ as per Bowtie2 computation of MAPQ for unireads. Let the minimum alignment score for reported alignments in Bowtie2 be represented as $s_{min} = -0.6 - 0.6 * (2 * read\ length)$. Then:

$$Z_{i,j} = \begin{cases} 42, & \frac{s_{i,j}}{s_{min}} \in [0, 0.2] \\ 40, & \frac{s_{i,j}}{s_{min}} \in (0.2, 0.3] \\ 24, & \frac{s_{i,j}}{s_{min}} \in (0.3, 0.4] \\ 23, & \frac{s_{i,j}}{s_{min}} \in (0.4, 0.5] \\ 8, & \frac{s_{i,j}}{s_{min}} \in (0.5, 0.6] \\ 3, & \frac{s_{i,j}}{s_{min}} \in (0.6, 0.7] \\ 0, & \frac{s_{i,j}}{s_{min}} \in (0.7, 1] \end{cases} \tag{11}$$

From this pseudo-MAPQ score, when conducting analyses in scored mode, we can compute the alignment quality $q_{i,j}$ as the probability that the alignment is aligned to the correct genomic locus from the definition of MAPQ as:

$$q_{i,j} = 1 - 10^{-z_{i,j}/10} \tag{12}$$

If the analysis is being run in unscored mode, each alignment quality $q_{i,j}$ is set to 1.

The set of alignment qualities associated with each read $r_i$ will be defined as $Q_i = \{q_{i,1} \ldots q_{i,k}\}$. We will define our initial weights $w_{i,j}$ by setting our initial prior probabilities $Pr(g_{i,j} = l_i)$ to be proportional to the alignment quality $q_{i,j}$. Because we assume that for each read $r_i$, one of the associated $g_{i,j} = l_i$, then for each read $r_i$, we set the associated alignment weights $w_{i,j}$ as:

$$w_{i,j} = \frac{q_{i,j}}{\sum_{q \in Q_i} q} \tag{13}$$

In scored mode, it is possible for the sum of the alignment qualities in $Q_i$ for a given read $r_i$ to be equal to zero; if this is the case, the read is discarded. Similarly, any alignments with alignment with a weight of zero are discarded. For all remaining reads and alignments, each weight $w_{i,j}$ is added to the appropriate chromosome dataset at the associated alignment locus $g_{i,j}$. Those reads with $k = 1$ are then removed from the list of reads over which to iterate because the weight off the associated alignment is fixed at $w = 1$.

When the initial assignment of prior probabilities as weights and addition of weights to the genome dataset is complete, then for each read $r_i$, the new weights can be computed as the posterior probabilities of $Pr(g_{i,j} = l_i \mid \text{total distribution of reads})$. First, we will represent the length of an alignment locus $g_{i,j}$ as $|g_{i,j}|$. Let $c_b$ be the sum of all weights associated with all alignments containing the genomic coordinate $b$. Then, we define $C_{i,j}$ as the average weight across the genomic coordinates of each alignment $g_{i,j}$ by the equation:

$$C_{i,j} = \frac{1}{|g_{i,j}|} \sum_{b \in g_{i,j}} c_b \tag{14}$$

Our algorithm assumes that the probability $Pr(g_{i,j} = l_i \mid \text{total distribution of reads})$ is proportional to $C_{i,j}$ and proportional to the alignment quality $q_{i,j}$. Based on this assumption, we define our likelihood function $F_{i,j}$ as the ratio of the average quality-weighted weight across the alignment locus to the weight of the alignment itself:

$$F_{i,j} = \frac{C_{i,j} q_{i,j}}{w_{i,j}} \tag{15}$$

By Bayes' theorem, we then state that our posterior probability is proportional to the likelihood and to the prior probability of a given event. To accommodate for slow fitting, we will define $r$ as the learning rate such that if $r = 0$, the weight will not change at all, and if $r = 1$, the new weight will be defined as per Bayes' theorem. When $r = 1$, then, we thus set our new weight $w'_{i,j}$ as our posterior probability $Pr(g_{i,j} = l_i \mid \text{total distribution of reads})$ by the equation:

$$w'_{i,j} = \frac{w_{i,j} F_{i,j}}{\sum_{j=1}^{k} w_{i,j} F_{i,j}} = \frac{C_{i,j} q_{i,j}}{\sum_{j=1}^{k} C_{i,j} q_{i,j}} \tag{16}$$

If $r$ is not equal to 1 (i.e. if fitting is conducted more slowly or faster), then per the above definition of the learning rate:

$$w'_{i,j} = \left( \frac{C_{i,j}q_{i,j}}{\sum_{j=1}^{k} C_{i,j}q_{i,j}} - w_{i,j} \right) r + w_{i,j} = \frac{rC_{i,j}q_{i,j}}{\sum_{j=1}^{k} C_{i,j}q_{i,j}} + (1-r)w_{i,j} \qquad (17)$$

Per this definition, when fitting is disabled (i.e. when $r = 0$), the new weight is not changed; when the fitting rate is set to $r = 1$, then Eq 17 is equal to Eq 16. Slower fitting can be achieved by setting $0 < r < 1$. The new weights are updated at the appropriate corresponding genomic loci, and the posterior weight $w'_{i,j}$ is treated as the prior weight $w_{i,j}$ for the next iteration. This process defined by Eqs 15–17 is conducted iteratively for the specified number of iterations.

After the specified number of iterations are complete, the output file is written by writing the prefix sum of the BIT $T_1$ for each chromosome at each position. If desired, the set of reads with corresponding weights are also written.

## Histone modification density and specificity computation

Because the ICeChIP-seq datasets have internal nucleosome standards bearing the targeted nucleosome modifications with uniquely identifying DNA "barcodes", we were able to calibrate our ChIP-seq results to yield the histone modification density (HMD), or the proportion of nucleosomes bearing the modification of interest. HMD for each dataset was computed as follows.

The average value of the BEDGRAPH for each of the calibrant barcodes was computed as above, and these values were grouped by the nucleosome modification associated with the barcode and summed, as previously described [29,30] for both the IP and the Input datasets. The ratio of the summed values for the targeted modification in IP over the same in Input was designated as the target enrichment $E_t$.

The HMD at each genomic locus was then computed as follows, where *IP* and *Input* represent the value of the IP and the Input at that genomic locus:

$$HMD(\%) = \frac{IP}{E_t * Input} * 100\% \qquad (18)$$

To generate genome-wide HMD BEDGRAPH files, the IP and corresponding Input genome coverage BEDGRAPH files outputted by the SmartMap software were merged with BEDTools unionbedg, and the HMD computation in (18) was used to compute HMD. Any region with an Input value of zero was set to an HMD of zero.

To compute the specificity, for those ICeChIP-seq datasets with calibrants bearing more than one modification with uniquely identifying DNA barcodes (AR9, AR16, and AR17), the enrichment of every species ($E_i$) was computed analogously to the $E_t$, and the specificity (as percent of target enrichment) was computed as:

$$Specificity(\%target) = \frac{E_i}{E_t} * 100\% \qquad (19)$$

## Alignment overlap analysis

To assess for overlap of alignments, we conducted SmartMap on the simulated dataset with the read weight output setting activated. Using bedtools intersect, we then identified alignments that intersected with the true read origin in the Gold Standard dataset. From this, by weight, we were able to compute three metrics. First, we computed the number of alignments by weight that were present in the intersected dataset as a proportion of the total number of

alignments by weight. Second, we computed the alignment weighted overlap proportion score, a measure of the proportion of a read's overall weight that overlaps with a given true origin of the read due to a given alignment. This is computed as the product of the weight of the alignment with the geometric mean of the proportion of overlap between the true read locus and the alignment locus. Finally, we computed the unweighted overlap proportion score, which is computed as the geometric mean of the proportion of overlap between the true read locus and the alignment locus.

### Repetitive element analysis

Repetitive elements for hg38 were obtained from the HOMER list of repeats [66]. The promoter was defined as the most upstream portion of the annotated repeat. This dataset was used for analyzing all repeats; for analyzing LINE elements, SINE elements, or Simple Repeats, the corresponding subset of the repeats was used.

The HMD profiles in Figs 6A and S10 were generated by computing the average HMD (from SmartMap analysis of AR16 and AR17) in 50bp windows from -1000bp to +1000bp relative to the promoter, with HMDs above 100% being set to 100% (because an HMD above 100% is definitionally impossible), and corresponding windows were averaged together to yield the average HMD profile for each set of elements.

To conduct clustering, first, the average HMD of the region -100bp to +100bp relative to each promoter in the relevant dataset was computed using the SmartMap analysis of AR16 and AR17, with HMDs above 100% being set to 100%. The data was then transformed to orthonormal basis by principal component analysis in R with scaling and centering. The resultant coordinate matrix used for k-means clustering, starting with 2 clusters and increasing the number of clusters until the decrease in total within-cluster sum of squares became markedly diminished; for each dataset (all repeats, LINE elements, SINE elements, and Simple Repeats), this occurred with 5 or more clusters and, accordingly, 4 clusters were used for each dataset.

RNA-seq analysis was conducted as follows. The average value of the RNA-seq SmartMap BEDGRAPH datasets were found across each LINE element. These values were then normalized to the SmartMap read count for each replicate (as average reads per million reads analyzed) and averaged to yield the average normalized read depth for each LINE element. These were then grouped by cluster and used to generate the quantile boxplots.

**Heatmap generation.**   Heatmaps of regions with nonzero HMD only in SmartMap analyses were generated as follows. The average HMD of the region -100bp to +100bp relative to each promoter was computed using both the uniread and SmartMap analyses of AR16 and AR17, with HMDs above 100% being set to 100%. Principal component analysis was conducted on the set of SmartMap HMDs in R with scaling and centering. Promoters with HMDs of zero in all of the uniread analyses and at least one nonzero SmartMap HMD were then selected and sorted by the first principal component. There were 142,392 such promoters. HMD profiles were then generated for each of the selected promoters as described above in 50bp windows from -1000bp to +1000bp relative to the promoter, but were not averaged together on corresponding windows. A field was added to the beginning of each row containing the value 100 as a calibration point for threshold adjustment.

The list of HMD profiles sorted on the first principal component was then imported into ImageJ as a Text Image. The height of the image was scaled down to 500pts with bilinear interpolation, and the thresholds were set from 0–100. The resultant image was exported as a PNG file, which was then opened in Photoshop in Indexed Color mode. The color table was then adjusted such that the lowest value was set to white and the highest value was set to the

appropriate color. The leftmost point of the image (corresponding to the added field with the calibration point value of 100) was then removed from the image to generate the final heatmap.

### Genome browser visualization

Genome browser visualization was conducted using Integrative Genomics Viewer (IGV) [67].

### Comparison to other methods

**Comparison to CSEM.** Comparison was attempted against the CSEM software for multi-read allocation [35] by only using the first read mate of our ICeChIP samples. However, the CSEM software returned a segmentation fault within the first minute of runtime, rendering comparison difficult.

**Comparison to BM-Map.** Comparison was attempted against the BM-Map software for multiread reweighting [36] by aligning the simulated read dataset with Bowtie2 per the settings used for SmartMap, followed by use of the BM-Map software with seven threads, the maximum permitted by the software. The first step of BM-Map (reading the alignments into memory) proceeded uneventfully, using one thread. However, shortly after the second step of BM-Map began, the software returned an error and exited without returning an output. This was observed with existing binaries and with compilation of the software from source. As such, we were unable to compare results from BM-Map to SmartMap.

**Comparison to iteration 0 and random alignments.** The simulated dataset was aligned with Bowtie2 per the settings used for SmartMap. The reads were then parsed to yield a single extended BED file as per SmartMapPrep. For the Random Alignment selection analysis only, the reads were then split into separate files based on the number of alignments per read, and the random_read_selection.R script from the SmartMap-analysis GitHub repository was used to randomly select one alignment per read. These datasets were then used in the SmartMap software with the score set to -60.6. For the iteration 0 dataset, the number of reweighting cycles was set to zero; for the Random Alignments analysis, the number of reweighting cycles was set to one.

**Comparison to Uniread.** The simulated dataset was aligned with Bowtie2 per the settings used for SmartMap, with the modification that no value was specified for the option -k. Uni-reads were then parsed from the output SAM file by selecting for reads with MAPQ scores of: 3, 8, 23, 24, 40, 42; these are the MAPQ scores that are assigned to unireads by Bowtie2 [40]. Reads were then parsed as per SmartMapPrep into a single extended BED file. This file was then used for SmartMap with one iteration, a minimum score of -60.6 and a maximum of one alignment per read.

### Statistical analyses

Statistical analysis for Fig 6C was conducted first with chi-square analysis on full contingency table and with post-hoc tests on collapse contingency tables as follows. For each of the datasets in Fig 6C, chi-square test for goodness-of-fit was conducted on the corresponding contingency table presented in Tables 4–6. The p-value for each of these tests was $p < 2.2x10^{-16}$ and, accordingly, post-hoc tests were conducted. The post-hoc tests consisted of collapsing each contingency table into a set of 2x2 contingency tables with the cluster of interest and family/type of interest compared to all other clusters and/or all other families within the contingency table. Chi-square goodness-of-fit tests were then conducted on each of these 2x2 contingency tables, and the p-values were Bonferroni corrected to adjust for the number of tests. These adjusted

p-values for each 2x2 contingency table test were used to label the graphs in Fig 6C as follows: $^*p<0.01$, $^{**}p<10^{-5}$, $^{***}p<10^{-10}$.

Statistical analysis on Fig 6D was conducted to compare median average normalized RNA-seq depth by cluster. Because the difference between cluster 3 and all of the other clusters appeared to be the most biologically meaningful, only pairwise comparisons were conducted between cluster 3 and the other clusters to limit the number of statistical comparisons and, accordingly, the degree of Bonferroni correction needed. Mood's median tests were solely conducted as pairwise comparisons between cluster 3 and each of the other clusters with Bonferroni correction to p values with n = 3 for Bonferroni correction. The adjusted p-values for each of these comparisons was $p<10^{-10}$ and was marked appropriately on the graph.

## Supporting information

**S1 Fig. Mappability of sampled loci and human genome. (A)** Number of regions from the true origin loci vs. average mappability (UMAP50) score of the loci. **(B)** Density of UMAP50 scores of 200bp windows across the human genome (hg38).
(TIF)

**S2 Fig. Characteristics of SmartMap with increasing iterations. (A)** Mean error of read depth at true origin loci in SmartMap scored mode vs. number of reweighting iterations. **(B)** Mean absolute error of read depth at true origin loci in SmartMap scored mode with a reweighting rate of 0.25 vs. number of reweighting iterations. **(C, D)** QQ plots of read depth in Gold Standard dataset vs. (C) uniread or (D) SmartMap (1 iteration) scored datasets. Color scale represents percentile of each point, from $1^{st}$ to $99^{th}$ percentiles. Dashed line represents line with slope of unity.
(TIF)

**S3 Fig. Validation and comparison of multiple mapping analysis. (A)** Average read depth of each dataset genome-wide. **(B)** Base pairs covered by MACS2 called peaks for each dataset. **(C)** Percentage of MACS2 peaks in the Gold Standard dataset intersecting with MACS2 peaks in each other analysis, as percentage of base pairs covered. **(D)** Percentage of MACS2 peaks in each analysis intersecting with MACS2 peaks in the Gold Standard dataset, as percentage of base pairs covered. **(E)** Average mean absolute error vs. mappability score (UMAP50) of each dataset. Dashed lines are presented for readability of overlapping curves rather than discontinuities in data. **(F)** Mean absolute error of read depth at true origin loci for each dataset, with Gold Standard as the reference point, stratified by average Gold Standard read depth at true origin locus. **(G)** Mean error of read depth at true origin loci for each dataset, with Gold Standard as the reference point, stratified by average Gold Standard read depth at true origin locus. **(H)** Mean unweighted overlap proportion between alignment and true read origin as a function of alignment weight for the no-iteration (green) and one-iteration (red) scored SmartMap analyses. Overlap proportion is computed as a geometric mean of the proportion of the alignment and of the true read origin that overlaps with the other.
(TIF)

**S4 Fig. SmartMap and uniread analyses of the 100bp read length validation dataset. (A, B)** Number of (A) alignments or (B) reads vs. number of alignments per read. **(C)** Quantile plot of read depth at the true origin loci. **(D)** Median read depth vs. mappability score (UMAP50) of the true origin loci. **(E-G)** QQ plot of read depth at true origin loci in the (E) SmartMap vs. uniread, (F) Gold Standard vs. uniread, and (G) Gold Standard vs. SmartMap scored datasets. Color scale represents percentile of each point, from 1st to 99th percentiles. **(H)** Mean absolute error of read depth at true origin loci for each dataset, with Gold Standard as the reference

point. **(I)** Average mean absolute error vs. mappability score (UMAP50) of each dataset.
(TIF)

**S5 Fig. Characteristics of the -k 101 SmartMap dataset. (A, B)** Number of (A) alignments or (B) reads vs. number of alignments per read. **(C)** Quantile plot of read depth at the true origin loci. Dashed lines are presented for readability of overlapping curves rather than discontinuities in data. **(D)** Median read depth vs. mappability score (UMAP50) of the true origin loci. **(E)** Mean absolute error of read depth at true origin loci for each dataset, with Gold Standard as the reference point. **(F)** Average mean absolute error vs. mappability score (UMAP50) of each dataset. **(G-J)** Average read depth across the bodies of (G) all repetitive elements, (H) LINEs, (I) SINEs, and (J) Alu elements.
(TIF)

**S6 Fig. Alignments per ICeChIP-seq dataset.** Number of alignments vs. alignments per read for each ICeChIP-seq dataset analyzed.
(TIF)

**S7 Fig. Reads per ICeChIP-seq dataset.** Number of reads vs. alignments per read for each ICeChP-seq dataset analyzed.
(TIF)

**S8 Fig. SmartMap and uniread analysis of AR8 input.** All analyses conducted on 200bp tiled genomic windows. **(A)** Quantile plot of read depth for SmartMap and uniread analyses. **(B)** Median read depth vs. mappability score (UMAP50) for SmartMap and uniread analyses. **(C)** Quantile plot of excess read depth in SmartMap relative to uniread analysis. **(D)** Median excess read depth vs. mappability score (UMAP50). **(E)** QQ plot of read depth in SmartMap vs. uniread analysis. Color scale represents percentile of each point, from $1^{st}$ to $99^{th}$ percentiles. Dashed line represents line with slope of unity.
(TIF)

**S9 Fig. SmartMap and uniread analyses of AR7, AR8, and AR9 HMDs. (A)** Mean or **(B)** Median HMD vs. mappability score (UMAP50) for SmartMap and uniread analyses. Red line represents SmartMap analysis; blue line represents uniread analysis.
(TIF)

**S10 Fig. Specificity scatterplots for AR9.** Scatterplots of (A) specificity or (B) log specificity for uniread vs. SmartMap analyses. Targets of pulldowns are H3K4me3 (left), H3K9me3 (centre), and H3K27me3 (right). Specificity is measured as the enrichment of each on- or off-target internal standard nucleosome as a percentage of on-target enrichment.
(TIF)

**S11 Fig. SmartMap analysis of ENCODE ATAC-seq datasets. (A-B)** Quantile plot of read depth at genomic windows in SmartMap and uniread analyses for (A) Replicate 1 or (B) Replicate 2. **(C)** Quantile plot of excess read depth in SmartMap datasets relative to corresponding uniread dataset for Replicates 1 and 2. **(D-E)** Median read depth vs. mappability score (UMAP50) in SmartMap and uniread analyses for (D) Replicate 1 or (E) Replicate 2. **(F)** Median excess read depth vs. mappability score (UMAP50). **(G)** Quantile plot of depth-normalized log ratio of read depth in Replicate 1 over Replicate 2, for SmartMap and uniread analyses. Graph breaks are present at both ends of the graph. **(H)** Mean absolute depth-normalized log ratio of the analyses shown in panel G.
(TIF)

**S12 Fig. SmartMap analysis of ENCODE RNA-seq datasets. (A-B)** Quantile plot of read depth at distinct Refseq genes in SmartMap and uniread analyses for (A) Replicate 1 or (B) Replicate 2. **(C)** Quantile plot of excess read depth in SmartMap datasets relative to corresponding uniread dataset for Replicates 1 and 2. **(D)** Quantile plot of depth-normalized distinct Refseq gene log ratio of read depth in Replicate 1 over Replicate 2, for SmartMap and uniread analyses. Pseudocount of $10^{-7}$ was added to each gene due to the high number of genes with zero read depth. Graph breaks are present at both ends of the graph. **(E)** Mean absolute depth-normalized log ratio of the analyses shown in panel D.
(TIF)

**S13 Fig. Histone modification and ATAC-seq profiles on subset clusters. (A-C)** HMDs of modifications about promoters of (A) LINEs, (B) SINEs, or (C) simple repeats separated by k-means clustering conducted on the appropriate set of repetitive elements. **(D-G)** Total ATAC-seq read depth across both replicates about promoters of (D) all repeats, (E) LINEs, (F) SINEs, or (G) simple repeats.
(TIF)

**S14 Fig. Heatmaps of repeat promoters under uniread analysis.** Heatmap of repeat promoters with measurable nonzero HMD only in SmartMap analysis, sorted on first principal component of repetitive elements.
(TIF)

## Acknowledgments

We would like to thank the ENCODE Consortium for providing the RNA-seq and ATAC-seq data used in this study. In particular, we thank the lab of Thomas Gingeras at Cold Spring Harbor Laboratory for generating the RNA-seq data, and we thank the lab of Michael Snyder at Stanford University for generating the ATAC-seq data.

## Author Contributions

**Conceptualization:** Rohan N. Shah.

**Data curation:** Rohan N. Shah.

**Formal analysis:** Rohan N. Shah.

**Funding acquisition:** Rohan N. Shah, Alexander J. Ruthenburg.

**Investigation:** Rohan N. Shah.

**Methodology:** Rohan N. Shah, Alexander J. Ruthenburg.

**Project administration:** Rohan N. Shah, Alexander J. Ruthenburg.

**Resources:** Alexander J. Ruthenburg.

**Software:** Rohan N. Shah.

**Supervision:** Rohan N. Shah, Alexander J. Ruthenburg.

**Validation:** Rohan N. Shah.

**Visualization:** Rohan N. Shah, Alexander J. Ruthenburg.

**Writing – original draft:** Rohan N. Shah.

**Writing – review & editing:** Rohan N. Shah, Alexander J. Ruthenburg.

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
