## [Decision Letter · Decision Letter 0]

13 Nov 2020

Dear Mr. Shah,

Thank you very much for submitting your manuscript "Sequence deeper without sequencing more: Bayesian resolution of ambiguously mapped reads" for consideration at PLOS Computational Biology.

As with all papers reviewed by the journal, your manuscript was reviewed by members of the editorial board and by several independent reviewers. In light of the reviews (below this email), we would like to invite the resubmission of a significantly-revised version that takes into account the reviewers' comments.

We cannot make any decision about publication until we have seen the revised manuscript and your response to the reviewers' comments. Your revised manuscript is also likely to be sent to reviewers for further evaluation.

Sincerely,

Ilya Ioshikhes

Associate Editor

PLOS Computational Biology

William Noble

Deputy Editor

PLOS Computational Biology

Reviewer's Responses to Questions

**Comments to the Authors:**

Reviewer #1: The authors present MultiMap, a method and software tool for aligning and reallocating multimapping reads. The core method uses bowtie2 set to report up to 51 alignments and then applies either a weighted (according to MAPQ) or unweighted EM procedure for a single iteration.

The C++ software implementing MultiMap is open source and available at: https://github.com/shah-rohan/multimap. I was able to build it with a minor hiccup (described in minor comments below).

The idea is far from new; there have been papers about this topic and suggesting substantially the same solution going back about 10 years (e.g. https://projecteuclid.org/euclid.cis/1268143264). That said, the authors do seem to be addressing a presently unmet need for methods that (a) handle-paired end reads, (b) are able to choose alignments for passing downstream, which is more widely applicable than quantifying a set of intervals, (c) handle strandedness. Thus, this method seems to be much more generic and likely to aid a variety of downstream tools than any other I know of.

Having said that the overall idea is not new, the idea of applying Fenwick trees to the bookkeeping is new and interesting. I am a little surprised this wasn't highlighted more as it's a good idea and I don't recall seeing it in any of the other papers describing a multimapper method.

The strongest point of the paper are the experiments, which are extensive and cover multiple downstream applications of multimapper reassignment.

Major comments

The authors could do more to investigate exactly why multimapper assignment performance suffers after the first iteration of reassignment. It should be easy to check the simulation results to see whether this is primarily because "real" peaks are pulling in reads not belonging to those peaks, or if it's for another reason.

There should be some discussion of the difference in computational efficiency between the uniread approach and MultiMap. Asking bowtie2 to report up to 51 alignments per read must have some measurable effect and it is important for readers to understand what it is.

Related to previous point: the paper should measure how much time and space overhead are added by the need to populate and query the Fenwick tree.

Minor comments

I recommend that the authors keep only source files and scripts in the repo and not pre-built libraries (.a files) or object files (.o files). I had to remove the .a and .o files from the gzstream subdirectory before I could successfully build the tool on my Mac.

The manuscript should tell readers what the UMAP50 score is. As of now, there just seems to be a reference to a paper, but I don't think it is a widely understood concept.

A very minor point: it is typical to say "gold standard" rather than "golden standard"

Reviewer #2: The authors present a new approach to handling multi-mapped reads, in particular for depth based analyses like ChIP-seq and RNA-seq. The approach works by reweighting MAPQ scores with all alternative mappings of a read and then iteratively reweighting MAPQ scores via the averaged weights of all reads mapping to the same locus.

# Major comments

1. From reading the manuscript, I am not entirely convinced that the approach to give reads at high coverage loci a bigger weight is correct in any case. I can imagine that there are loci which (wrongly) attract a lot of reads from elsewhere in the genome. With this approach, it seems to me that reads mapping at such loci will get an artificially good weight, thereby even increasing the systematic error that the read mapper made there. How does the method protect against such a scenario, or alternative, why is it likely to be not a problem?

2. The manuscript lacks a comparison against the other cited methods. I understood that they do less and CSEM fails, but the others could still be applied in order to illustrate how MultiMap improves over them.

3. As I understood it, the reweighting of the reads could be used to provide basically improved MAPQ values (see Discussion), i.e. a better estimate of the probability that a read (pair) alignment is mapped to the wrong place. With the simulated golden standard, it should be easy to actually calculate whether the new updated MAPQ scores are correct, or at least better than the initial MAPQ scores from the mapper. Among all read (pairs) with an estimated MAPQ probability p, there should be only p wrongly mapped. The true fraction of wrongly mapped read pairs should be calculated for each p, and plotted against the values of p. If the algorithm works well, the result should be nearly a diagonal. Such an analysis would generate much more trust in the used weighting than the indirect approaches done here (although they are of course also very reasonable).

4. Fig S2A is unclear. Why is the depth depending on number of iterations not just compared in a qq-plot against the golden standard depth as in S2B and S2C?

5. It might be just me, but I don't understand how the Golden Standard can have an off-target read depth (Fig 3G). Isn't Golden Standard the true depth at each locus?

6. Fig 3J should probably be stratified by locus read depth. This way, one could see whether the multiMap approach generates some kind of biased errors towards high depth loci (see 1.).

7. For the sake of reproducibility, the authors should provide their data analysis code along with the paper, e.g. by citing a Zenodo archive or a git repository. Ideally, the analysis would be implemented in a fully reproducible, automated way (e.g. using a workflow management system).

8. The golden standard dataset was generated with 50bp read length. Given current habits, this is an artificially low length, which probably artificially magnifies the advantage of multiread compared to uniread analysis. I think, to get a more up to date picture, the golden standard analysis should be repeated with 100bp reads.

9. The MultiMap software on Github definitely needs a Github release with proper versioning. Ideally, it should also be made available via Bioconda (but that is just a suggestion as I am biased in this regard).

10. The authors say that their reweighting is a Bayesian approach. I can see from the formulas that calculations resemble Bayes theorem, however the entire presentation is too ad hoc. The method presentation definitely needs an accurate definition of involved observed and latent variables, their modeled distributions and an explicit definition of their conditional dependencies. If the obtained weights are meant to be probabilities, please also define them as such, using the usual notation.

# Minor comments

10. Fig 3: the red and the orange are hard to distinguish.

11. Fig 3E: How is excess read depth defined?

12. Fig S2F: why are there gaps in the curve for M1S?

13. The performance comparison against the other method at line 554 seems quite selective (chosing just one method) and does not even run both methods on the same machine. To really make a statement about performance, this has to be done in a comparable setting on the same dataset. If the authors claim that their tool is performing best, the have to show it properly comparing against all others, on the same machine and data.

Johannes Köster

Reviewer #3: In this paper the authors present a new approach and software for reweighting ambiguously mapped reads in NGS data, including ChIP-seq, ATAC-seq, and RNA-seq. Their approach uses binary indexed trees to store map counts and use an iterative algorithm to reweight alignments based on quality scores.

The primary application of this approach appears to be for “peak-type” NGS data, such as ChIP-seq and ATAC-seq, rather than RNA-seq. MultiMap does not directly output count data and thus does not easily fit into standard RNA-seq workflows. Also, reported challenges with gapped/spliced alignments would also disqualify MultiMap from RNA-seq analysis. However, the emphasis on peak data is actually an advantage for MultiMap since the ambiguous mapping problem in peak data has received far less attention than in RNA-seq (more on this below). Given the number of RNA-seq disambiguation approaches that have been proposed to date, the authors may consider focusing their tool on peak data and downplaying (or even removing) the RNA-seq applications.

The most novel aspect of their approach is that there is no need for a priori assumptions about genomic features of interest (lines 99-102). This sets MultiMap apart from every RNA-seq disambiguation approach. I think further emphasis of this point would help to distinguish their approach from existing approaches.

A major weakness of this paper is that authors do not seem to be current on the relevant literature. The past 5 years has seen an explosion of new approaches for solving the problem of ambiguously mapped reads in RNA-seq. The review article by Lanciano and Cristofari (doi: 10.1038/s41576-020-0251-y) lists over a dozen approaches with this specific purpose.

Specific comments:

1. A thorough literature search would properly frame the importance and novelty of this work.

a. Line 96 - Should cite the review by Lanciano and Cristofari and the approaches therein.

b. Line 157 - There are previous works that use paired-end sequencing information and alignment quality.

2. This study is missing a rigorous performance comparison with existing methods – the only method compared was unireads. If

MultiMap is intended to be applied to RNA-seq data, it needs to be compared to RNA-seq methods (see Lanciano and Cristofari).

3. This study makes claims of improved computational efficiency but does not compare to existing approaches. Unireads approach (discarding ambiguous reads) will certainly be more computationally efficient.

4. I am not convinced that 51 alignments per read is sufficient. Some Alu families number in the tens of thousands of insertions and (depending on read length) may map hundreds of times. The peak at 51 in Fig 2 suggests that some reads have many more alignments and -k is too low.

5. MultiMap algorithms, including Fenwick trees and iterative model, seem to be the same as presented in Chung et al. (doi: 10.1371/journal.pcbi.1002111). The main improvements appear to be support for paired reads and allowing for non-fixed number of iterations.

6. “Overfitting” may not be the proper term to describe deviation from the gold standard after the first iteration, since you are not adding parameters to the model. It is also interesting that your algorithm converges after a single iteration – it suggests that your model places too much weight on the (error-prone) observed data which may be incorrect due to sequencing error or mismatches with the reference. One way to mitigate this is to use an informative prior that assigns initial non-zero weights that are reweighted by the observed data.

7. Line 384: “impute” is used incorrectly

**Have all data underlying the figures and results presented in the manuscript been provided?**

Reviewer #1: Yes

Reviewer #2: Yes

Reviewer #3: Yes

PLOS authors have the option to publish the peer review history of their article (what does this mean?). If published, this will include your full peer review and any attached files.

Reviewer #1: No

Reviewer #2: **Yes: **Johannes Köster

Reviewer #3: No
---

## [Decision Letter · Decision Letter 1]

1 Mar 2021

Dear Mr. Shah,

Thank you very much for submitting your manuscript "Sequence deeper without sequencing more: Bayesian resolution of ambiguously mapped reads" for consideration at PLOS Computational Biology.

As with all papers reviewed by the journal, your manuscript was reviewed by members of the editorial board and by several independent reviewers. In light of the reviews (below this email), we would like to invite the resubmission of a significantly-revised version that takes into account the reviewers' comments.

We cannot make any decision about publication until we have seen the revised manuscript and your response to the reviewers' comments. Your revised manuscript is also likely to be sent to reviewers for further evaluation.

Sincerely,

Ilya Ioshikhes

Deputy Editor

PLOS Computational Biology

William Noble

Deputy Editor

PLOS Computational Biology

Reviewer's Responses to Questions

**Comments to the Authors:**

Reviewer #1: This revision has addressed my comments. While there are places where I think the language could use work ("as the old saying goes" is a tad colloquial for a technical paper but I defer to the editor :-)), and some of the figures look blurry to me, I think the reversion is overall extremely thorough and the paper is an exciting contribution.

Reviewer #2: Most of my requests were properly adressed. Two points remain below.

# Major comments

* 2.7: I am glad to see the scripts being online in the github repo. Similar to the software, this repo as well needs a release that is properly cited in the paper. Apart from that, the analysis is still not reproducible. The repository lacks a readme, that would explain me how to run the analysis. It seems to lack steps that obtain the necessary reference data, and simulate the reads. It lacks the definition of a software environment to use (e.g. a Conda environment or a container image).

# Minor comments

* Comment on the authors response to comment 1.5: I've still found the wording "golden standard" in Figure S2.

Johannes Köster

Reviewer #3: The authors adequately addressed all concerns and the new manuscript is much improved. The decision to position this tool mainly for “peak” data effectively solved many of the problems with its applications to transcriptome data.

I agree, it is very interesting that one iteration still gives the lowest error, and even with the slow-reweighting algorithm. I would be curious whether situations exist where multiple iterations would lower the error, though not necessary for this study.

Minor suggestions

Line 75-76: It is widely accepted that repetitive elements comprise >50% of the human genome. Typical citations establishing this include Lander et al. 2001 (initial human genome sequence), Wheeler et al. 2013 (Dfam), etc.

Line 191: writ -> at

Line 811-812: It has less to do with “poor quality” alignment as it has to do with the repetitiveness of the genome. Remember plants i.e. maize can have up to 90% repetitive DNA. Different genomes may require different -k values for optimal tradeoff between usable and discarded reads.

**Have all data underlying the figures and results presented in the manuscript been provided?**

Reviewer #1: Yes

Reviewer #2: **No: **The workflow code lacks a way to obtain reference data and software, as well as the simulation code.

Reviewer #3: Yes

PLOS authors have the option to publish the peer review history of their article (what does this mean?). If published, this will include your full peer review and any attached files.

Reviewer #1: No

Reviewer #2: **Yes: **Johannes Köster

Reviewer #3: No
---

## [Decision Letter · Decision Letter 2]

23 Mar 2021

Dear Mr. Shah,

Thank you very much for submitting your manuscript "Sequence deeper without sequencing more: Bayesian resolution of ambiguously mapped reads" for consideration at PLOS Computational Biology. As with all papers reviewed by the journal, your manuscript was reviewed by members of the editorial board and by several independent reviewers. The reviewers appreciated the attention to an important topic. Based on the reviews, we are likely to accept this manuscript for publication, providing that you modify the manuscript according to the review recommendations.

Sincerely,

Ilya Ioshikhes

Deputy Editor

PLOS Computational Biology

William Noble

Deputy Editor

PLOS Computational Biology

[LINK]

Reviewer's Responses to Questions

**Comments to the Authors:**

Reviewer #2: I am happy to see the analysis being described quite comprehensively, thanks a lot for making this effort. I am also glad to see that not only the future readers but even the authors themselves even had a benefit from this, by discovering bugs in the previous analysis.

Reviewer #3: The authors have done a thorough job addressing the reproducibility concerns by including additional documentation and example data sets.

Suggestions:

Note that MultiMap is the name of an existing computational genetics software (PMID:8054979) with 109 citations.

I do think the point that Dr. Köster is making is valid – if your goal is to produce software that can be used by others, there is still more that can be done. When I see a new software that has 10 dependencies that need to be installed, I usually just look for another package. Best practices would be to provide a way that dependencies can be easily managed, and a package manger (like conda/bioconda) is one of the easiest ways to do this. With conda you simply list these dependencies in a YAML file and conda takes care of resolving them.

Simulation data can be more easily made available by providing the simulator command and specifying a random seed. Since simulations are relatively inexpensive to generate but can easily produce terabytes of data, it is much more effective to provide simulation scripts that can precisely recreate your simulations. Most simulators will reproduce the exact simulated sequence by specifying a random seed; see “-seed” for bedtools and “--rng” for NEAT.

**Have all data underlying the figures and results presented in the manuscript been provided?**

Reviewer #2: Yes

Reviewer #3: Yes

PLOS authors have the option to publish the peer review history of their article (what does this mean?). If published, this will include your full peer review and any attached files.

Reviewer #2: **Yes: **Johannes Köster

Reviewer #3: No

Figure Files:

Data Requirements:

Reproducibility:

References:

If you need to cite a retracted article, indicate the article’s retracted status in the References list and also include a citation and full reference for the retraction notice.

---

## [Decision Letter · Decision Letter 3]

30 Mar 2021

Dear Mr. Shah,

We are pleased to inform you that your manuscript 'Sequence deeper without sequencing more: Bayesian resolution of ambiguously mapped reads' has been provisionally accepted for publication in PLOS Computational Biology.

Best regards,

Ilya Ioshikhes

Deputy Editor

PLOS Computational Biology

William Noble

Deputy Editor

PLOS Computational Biology

Reviewer's Responses to Questions

**Comments to the Authors:**

Reviewer #3: Excellent effort in making your tool more user friendly and addressing all suggestions.

**Have all data underlying the figures and results presented in the manuscript been provided?**

Reviewer #3: Yes

PLOS authors have the option to publish the peer review history of their article (what does this mean?). If published, this will include your full peer review and any attached files.

Reviewer #3: No

---

## [Editor Report · Acceptance letter]

14 Apr 2021

PCOMPBIOL-D-20-01727R3 

Sequence deeper without sequencing more: Bayesian resolution of ambiguously mapped reads

Dear Dr Shah,

I am pleased to inform you that your manuscript has been formally accepted for publication in PLOS Computational Biology. Your manuscript is now with our production department and you will be notified of the publication date in due course.

With kind regards,

Katalin Szabo
